# Immune signature of Chlamydia vaccine CTH522/CAF®01 translates from mouse-to-human and induces durable protection in mice

Anja W. Olsen [1] ✉, Ida Rosenkrands [1], Christina S. Jacobsen [1,3], Hannah M. Cheeseman [2], Max P. Kristiansen [1], Jes Dietrich [1], Robin J. Shattock [2] & Frank Follmann [1]

The clinical development of an effective Chlamydia vaccine requires in-depth understanding of how well protective pre-clinical immune signatures translate to humans. Here, we report a comparative immunological characterization of CTH522/CAF®01 in female mice and humans. We find a range of immune signatures that translate from mouse to human, including a Th1/Th17 cytokine profile and antibody functionality. We identify vaccine-induced T cell epitopes, conserved among Chlamydia serovars, and previously found in infected individuals. Using the mouse model, we show that the common immune signature protected against ascending infection in mice, and vaccine induced antibodies could delay bacterial ascension to the oviduct, as well as development of pathology, in a T cell depleted mouse model. Finally, we demonstrate long-lasting immunity and protection of mice one year after vaccination. Based on the results obtained in the present study, we propose to further investigate CTH522/CAF®01 in a phase IIb study.

At present, no Chlamydia vaccine has reached the market. Predicting human immune responses on the basis of animal data can be challenging, as species specific responses to the antigens/adjuvants, are not uncommon[1]. In the Chlamydia field CTH522 formulated in CAF®01 is the first vaccine that has passed phase I clinical trial with a satisfactory safety profile and primary endpoints (immunogenicity by seroconversion) being reached[2]. This gave us a unique opportunity to perform a comparative analysis with a Chlamydia vaccine, in order to identify to what extent mouse data can be translated to humans.

Sexually transmitted *Chlamydia trachomatis (C.t.)* is a common urogenital infection affecting both men and women[3]. As a significant number of cases are asymptomatic and consequently untreated[4], infections can lead to serious complications, including pelvic inflammatory disease (PID), fibrotic blockage of the oviducts and resulting infertility or ectopic pregnancy[5]. In general, data from animal models point to a protective role of both cell mediated immune (CMI) responses[6], antibodies[7–10], and the interplay between the two[10,11]. In humans, correlates of protection against *C.t.* infection, reinfection, and ascending infection have been associated with Th1 responses and IFN-γ secretion[12–15], whereas the role of antibodies is still being debated. Studies have failed to detect a clear role for infection induced antibodies in controlling an ascending infection[16] and increased levels of anti-chlamydia IgG have been associated with higher risk for incident infection[17] and disease[18]. On the other hand, local IgA[19] and serum IgG[16] levels correlate with reduced bacterial burden at the cervix.

In our vaccine research program, we have been dedicated to the design of vaccine constructs with the ability to induce surface recognizing and neutralizing antibodies in combination with a strong CMI response. The Major Outer Membrane Protein (MOMP) of *C.t.* has several key antigenic features needed for protective immunity against

[1]Center for Vaccine Research, Department of Infectious Disease Immunology, Statens Serum Institut, Copenhagen, Denmark. [2]Department of Infectious Disease, Imperial College London, London, UK. [3]Present address: PharmaRelations, Virum, Denmark. ✉e-mail: Aol@ssi.dk

infections, via a combination of surface targeting B cell epitopes and conserved T cell epitopes[10,20]. A range of studies using both the *C. muridarum*[21–23] and *C.t.*[24] mouse challenge models, have demonstrated protection with MOMP based vaccines. CTH522 is composed of the majority of the MOMP sequence from SvD (aa34-aa259), a sequence that holds highly conserved regions and is known to contain T and B cell epitopes[25]. This region is followed by a repeat of extended Variable Domain 4 (VD4) regions from SvD, E, F, and G, designed to elicit broadly cross-reactive immune responses able to neutralize multiple serovars[10,26], representing up to 90% of the human *C.t.* prevalence[27]. Initial pre-clinical evaluation of CTH522 formulated in the Th1/Th17 inducing CAF®01 adjuvant showed that the vaccine was highly immunogenic and protective after both vaginal[10] and transcervical infections[28].

For the phase I clinical trial of CTH522, a GMP production of CTH522 was performed in *E. coli*. Following purification procedures, CTH522 showed an unusual self-assembly into nanoparticles, a feature that may be favorable with regard to immunogenicity of the protein[29]. In the current study, we investigated immune signatures of the CTH522 GMP product, formulated in cationic adjuvant formulation CAF®01 in pre-clinical mouse models and studied whether they were translatable to humans. Firstly; we performed a comparative immunological characterization of CTH522/CAF®01 vaccinated mice and humans, secondly; we investigated the in vitro and in vivo protective effect of murine and human antibodies, including additional data to support a role for CTH522 specific antibodies in controlling the ascending infection and thirdly; we evaluated whether this immune signature led to sustained protection evaluated by long-term (1 year) immunogenicity and protection in the mouse model.

Here, we show that CTH522 specific immune responses generated in the mouse model are, to a high degree, translatable to humans regarding both the specificity and functionality of the antibody response (surface recognition of prevalent serovars, IgG subclass profile, surface exposed B cell epitopes, neutralizing capacity) and the cytokine profile (Th1/Th17 responses) of the CMI response. Moreover, T cell epitope mapping identify T cell epitopes in conserved regions for both species, and for humans the T cell epitope regions overlap known infection driven T cell epitopes. This common immune signature leads to protection against ascending infection and pathology in mice and is sustained and protective for up to 1-year post vaccination. Overall, the outcomes of our comparative analysis of CTH522 induced immune responses in mice and humans are promising in relation to the protective efficacy of CTH522 as a Chlamydia vaccine for humans, that is likely to be introduced alongside the HPV vaccine and therefore needs to be sustained for a least a decade.

## Results

### Structure of CTH522 and experimental outlines
CTH522 is a recombinant engineered version of *C.t.* MOMP. MOMP holds four highly variable domains VD1 to VD4, spaced between serovar conserved segments (CSI-CSV)[30]. CTH522 is composed of the majority of the MOMP sequence from SvD (aa34-aa259; termed "CTH523"), a sequence that holds highly conserved regions among serovars and known to contain human T and B cell epitopes[25]. CTH523 is placed end to end with a repeat of extended VD4 regions from SvD to SvG (termed "CTH518") (Fig. 1a), designed to elicit a broadly cross-reactive immune response. A comparative evaluation of T and B cell responses in mice and humans after vaccination with CTH522 formulated in CAF®01 was undertaken followed by a thorough characterization of the ability of the vaccine to protect against a vaginal challenge short- and long-term post vaccination (Fig. 1b).

### Th1/Th17 T cell responses and T cell epitope mapping
T cell responses in CTH522 vaccinated mice and humans were assessed with the overall aim to localize T cell epitopes in CTH522 and inform on

the potential coverage of the CMI response against multiple serovars. B6C3F1 mice were vaccinated with 10 µg CTH522 formulated in CAF®01. PBMC samples from participants vaccinated with CTH522/ CAF®01 were obtained from the first-in-human clinical trial (CHLM-01)[2]. In mice, the immune profile of MOMP formulated with CAF®01 has previously been described in ref. 31 and the dominating cytokines are IFN-γ and IL-17A, together with IL-6 and TNF-α. Here, we found high levels of IFN-γ (median levels of ~46.000 pg/ml) and prominent levels of IL-17A (median levels of ~900 pg/ml) cytokines in supernatants of CTH522 stimulated mouse splenocytes (Fig. 1c). Similarly, we characterized the CMI responses in the CHLM-01 participants and in agreement with the mouse data, both IFN-γ and IL-17A were present in the supernatants of CTH522 stimulated PBMCs (Fig. 1d) together with IL-6, TNF-α, and IL-13 as the dominating cytokines (Supplementary Fig. 1). To demonstrate that CTH522 specific T cells could recognize *C.t.* bacteria, mouse splenocytes were stimulated in vitro with UV-inactivated *C.t.* SvD. Strong recognition of *C.t.* SvD was found after CTH522 vaccination (Supplementary Fig. 2).

To localize and compare the breadth of the T cell epitope regions of CTH522, mouse splenocytes and human PBMCs were further stimulated in vitro with CTH523 and CTH518 (Fig. 1e, f). Both regions of CTH522 induced a T cell response leading to a more detailed epitope mapping of the whole CTH522 sequence (Fig. 1g). Using individual 20–21mer (10aa overlap) peptides covering the CTH522 sequence (Fig. 1g and Supplementary Table 1 for sequences) the main T cell epitopes in B6C3F1 mice were identified to regions covering P8[D], P16-17[D], P25[D] and sequences in and around P30 from the four serovars (Fig. 1g left *y*-axis, red bars). Peptides P16-17[D] and P25[D] are localized in the conserved segments CSII and CSIV, respectively. P8[D] is localized in the transition between VD1 and CSII and finally P30 is localized within the VD4 region, and comprises the highly serovar conserved TTLNPTIAG sequence. Besides these epitopes, serovar specific epitopes were identified in P28–P29[D, E] and P29[F]. To identify human T cell epitopes, PBMC samples with a total CTH522 specific IFN-γ response >500 pg/ml after background subtraction were selected (*n* = 10). The results demonstrated that many of the peptides were able to induce IFN-γ secretion from one or more of the CTH522 responders (Fig. 1g, right axis, blue bars and Supplementary Fig. 3 for response levels). Some of the T cell epitope regions were identified in both mice and humans; P16-17[D], P25[D] and P30. However, the most frequently recognized regions, recognized by >50% of the CTH522 responders; P10[D] (CSII), P19[D]–20[D] (CSIII), P28[F] and P28[G] (CSIV) were not identified as strong T cell epitope regions in mice. Across a broad range of *C.t.* serovars the regions covering P10[D] (CSII) and P19[D]–20[D] (CSIII) were 95–100% conserved, P28[F] 70–80% and P28[G] 75–85% (Supplementary Table 1). This is very promising in relation to cross-serovar protection. Importantly, some of these regions have also been identified as T cell epitope regions in humans infected with *C.t.*[25,32,33]. Overall, we demonstrated that CTH522 holds strong T cell epitopes in both conserved and variable regions of CTH522 in both mice and humans, that the strong human T cell epitopes in the more conserved regions of MOMP were highly conserved across serovars, and finally that we see overlap with known epitopes recognized during a *C.t.* infection in humans.

### Antibody surface recognition, subclass distribution, and B cell epitope mapping
CTH522 was designed to induce a strong antibody response recognizing the surface of the prevalent sexually transmitted serovars (D-G). A characterization of the antibody response after CTH522/CAF®01 vaccination demonstrated strong recognition of the bacterial surface of *C.t.* SvD, E, F, and G in both mice and humans (Fig. 2a, b). As antibodies can have multiple effector functions involving Fc regions on specific IgG subclasses, we next investigated the presence of different IgG subclasses. Both IgG1, IgG2a/IgG2c,

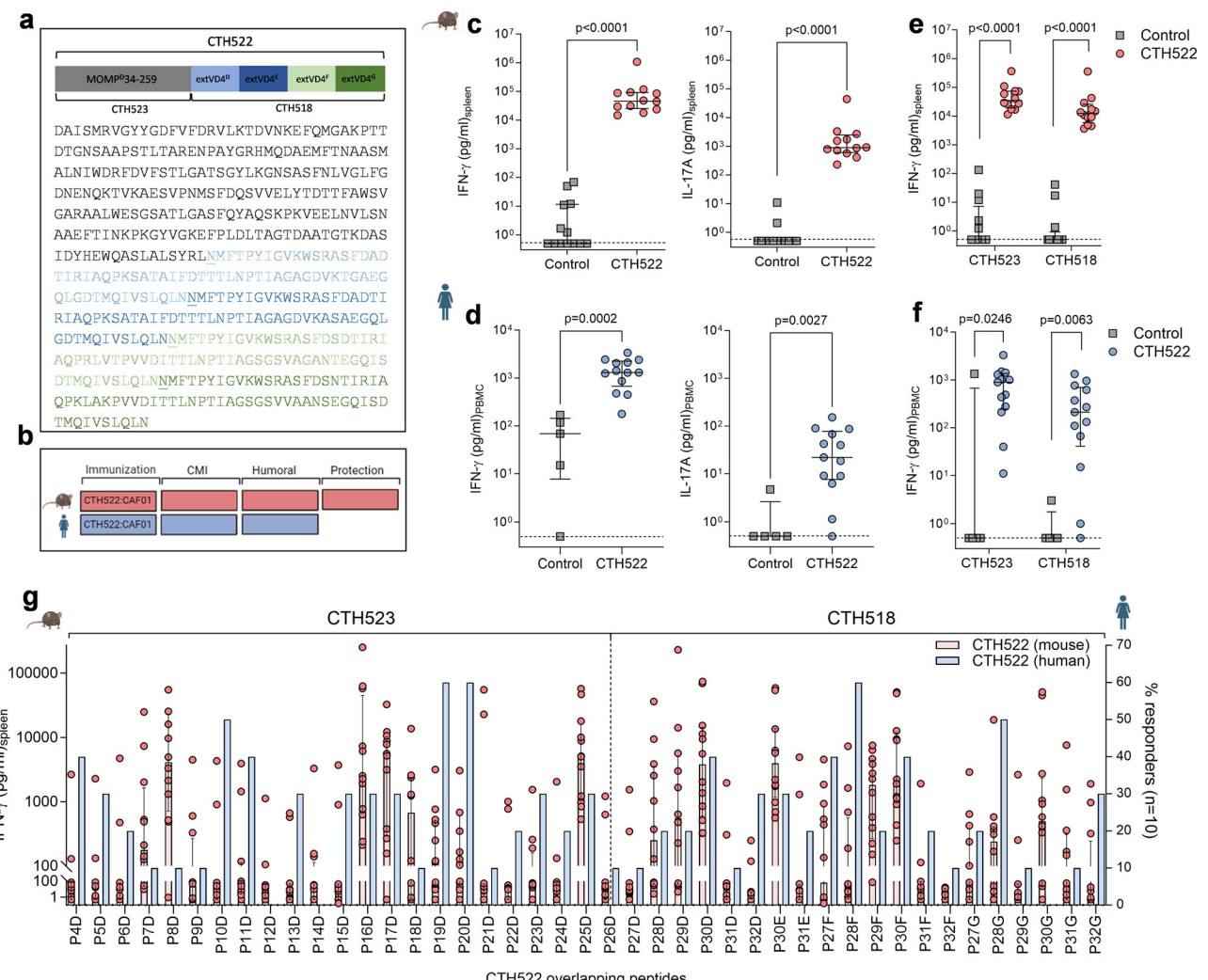

**Fig. 1 | T cell responses after CTH522/CAF®01 immunization in mice and humans. a** Design and sequence of CTH522, CTH523 and CTH518. **b** Experimental outline. **c**, **e**, **g** T cell responses in vaccinated mice. Female B6C3F1 mice were immunized three times s.c. with 10 μg CTH522/CAF®01 (CTH522) or sham-immunized (Control). Two weeks post 3rd immunization individual splenocytes were isolated and stimulated in vitro with (**c**) CTH522, (**e**) CTH523 and CTH518. IFN-γ and IL-17A responses were measured in supernatants after 3 days by ELISA. Graphs represent a pool of three individual experiments (sham-immunized $n = 13$, CTH522 immunized $n = 12$). CTH522/CAF®01 vaccinated and placebo participants. PBMC samples from CTH522 vaccinated ($n = 13$) and placebo samples ($n = 5$) were stimulated in vitro with (**d**) CTH522 and (**f**) CTH523 and CTH518. IFN-γ and IL-17A responses were measured in supernatants after 5 days by ELISA. Data points for both mice and humans represent cytokine levels in antigen stimulated wells after subtraction of media only and lines represent median responses with 25th and 75th percentiles for each group. **c**–**f** Two-tailed Mann–Whitney $U$ test was used for comparison among groups. **g** IFN-γ responses after in vitro stimulation with 20–21-mer peptides overlapping CTH522 (Supplementary Table 1). Left $y$-axis (red bars) splenocytes from CTH522 vaccinated mice ($n = 12$). Each bar shows median peptide specific IFN-γ responses with 25th and 75th percentiles. Right $y$-axis (blue bars) peptide responses of human PBMC samples with a total CTH522 IFN-γ response >500 pg/ml ($n = 10$). Blue bars represent % responders out of the 10 donors with individual peptide responses >170 pg/ml (mean of peptide responses in the placebo group +4*SD). **b** Figure and **c**–**g** Mouse and human symbols were created with Biorender.com. **c**–**g** Source data are provided as a Source Data file.

and IgG2b were present in CTH522 specific serum from mice (Fig. 2c). In humans IgG1 and IgG3 dominated the response with low levels of IgG2 and IgG4 (Fig. 2d)−a subclass distribution, that in both mice and humans, enables a variety of Ab effector functions. A detailed B cell epitope mapping, using the synthetic peptide approach used for T cell epitope mapping, demonstrated that the B cell epitopes were localized to surface exposed regions of MOMP (Fig. 2e, mice (red bars, left y-axis) and humans (blue bars, right y-axis)). The VD4 regions represented by P30, comprising the highly conserved and neutralizing TTLNPTIAG region were together with regions surrounding VD1 (P6 and P8), strong B cell epitope regions in both mice and humans. In humans, in contrast to mice, we further identified a region in close proximity to VD3 (P21-23) to hold human B cell epitopes. See Supplementary Fig. 4 for peptide responses in control samples.

Having detected strong surface recognition of all four serovars in both mice and humans combined with recognition of the neutralizing VD4 regions (P30^D, P30^E, P30^F, P30^G) we next investigated the ability of CTH522 specific antibodies to neutralize SvD, E, F, and G. CTH522 specific serum from both mice and humans were able to neutralize all four serovars (NT$_{50}$ = reciprocal 50% neutralization titer) although, neutralization of SvF was slightly lower compared to the three other serovars (Fig. 2f, g). The VD4 region has been demonstrated to be surface exposed on B-complex, B-related and C-related complex serovars, but not on C-complex serovars[34]. To further test the ability of CTH522 specific serum to cross-neutralize serovars not represented in the vaccine construct, a pool of mouse serum with a SvD NT$_{50}$ ≈ 5000 and a pool of human serum with SvD NT$_{50}$ ≈ 500 were tested for its ability to cross-neutralize SvH, I, Ia, J and K and ocular SvA, B, and C. We found that high titer CTH522 specific mouse serum was

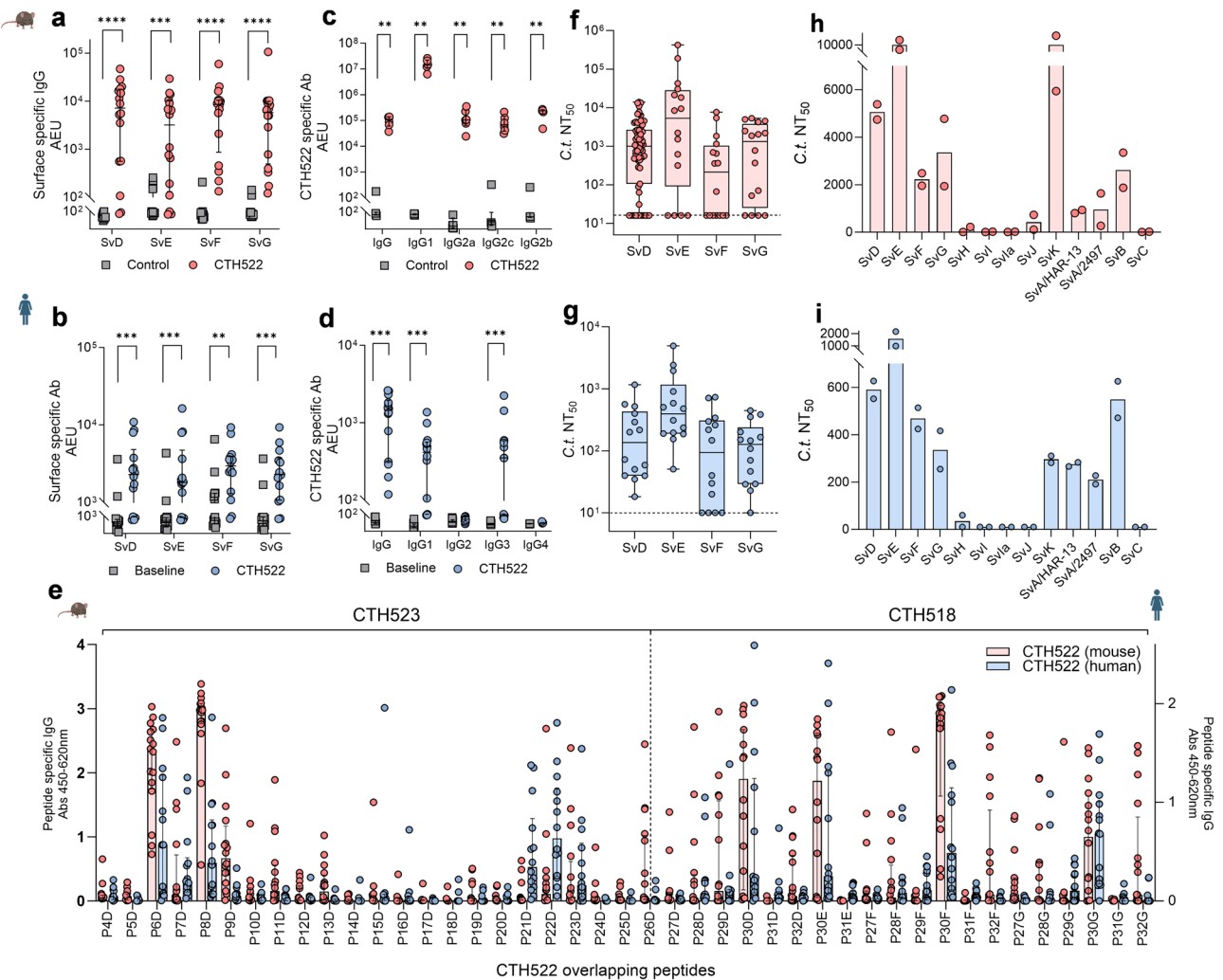

**Fig. 2 | Characterization of antibody responses in CTH522 vaccinated mice and humans. a–d** Antibody responses in CTH522/CAF®01 vaccinated and control B6C3F1 mice and in CTH522/CAF®01 vaccinated humans and baseline samples. Data points are presented as arbitrary Elisa Units (AEU) and represent IgG levels for individual serum samples and lines represent median responses with 25th and 75th percentiles for each group. **a, b** *C.t* SvD, E, F, and G specific IgG. **a** Mouse samples (*n* = 16). Two-tailed Mann–Whitney *U* test was used for comparison, ****$p$ < 0.0001, ***$p$ = 0.0005. **b** Human samples (*n* = 14). Two-tailed Wilcoxon matched-pairs signed rank test was used for comparison, SvD and SvE ***$p$ = 0.0002, SvF **$p$ = 0.0012, SvG ***$P$ = 0.0004. **c, d** IgG subclass distribution. **c** Mouse samples (*n* = 6). Two-tailed Mann–Whitney *U* test was used for comparison among groups **$p$ = 0.0022. **d** Human samples (*n* = 14). Two-tailed Wilcoxon matched-pairs signed rank test was used for comparison among groups IgG, IgG1, IgG3 ***$p$ = 0.0001. **e** B cell epitope mapping in serum samples of CTH522/CAF®01 vaccinated mice and

humans. Mouse (left *y*-axis, red bars, *n* = 16) and human (right *y*-axis, blue bars, *n* = 14) serum samples were diluted 1:200 and the fine specificity of the IgG responses (OD values) was studied using CTH522 specific overlapping peptides. Each bar represents the median response against each peptide with 25th and 75th percentiles. **f, g** In vitro neutralization of *C.t.* SvD, E, F, and G with serum samples from CTH522/CAF®01 vaccinated mice and humans. **f** Mouse samples SvD, *n* = 67, SvE to G, *n* = 16. **g** Human samples SvD, E, F, and G, *n* = 14. Data points for both mice and humans represent the calculated mean NT50 of duplicate or triplicate readings and are presented as Box and Whiskers plots with median and 25th and 75th percentiles (Box) and Min to Max, all data points (Whiskers). **h, i** NT50 against *C.t.* Sv's (**h**), mouse serum pool and (**i**), human serum pool. Data points represent the calculated mean NT50 of duplicate or triplicate readings and each bar shows the mean of two individual determinations. Mouse and human symbols were created with Biorender.com. **a–i** Source data are provided as a Source Data file.

able to cross-neutralize SvB and SvK and to a lesser extent SvA (represented by two different strains), but only minor or no cross-neutralization of the C-complex serovars SvC, H, I, Ia, and J were detected (Fig. 2h). Similar results were found with the human serum pool (Fig. 2i). Contrary to this, the C-complex serovars (SvA, C, I, Ia, and J) and also the C-related SvK were strongly neutralized with an immuno-repeat construct based on an extended VD1 region from SvIa (extVD1Ia*4) (Supplementary Fig. 5). These results are in line with previously published result using immuno-repeat constructs based on extended VD1 regions from C-complex serovars (SvA and SvJ)[35].

In order to demonstrate that CTH522 specific IgG was the isotype responsible for the ability to reduce *C.t.* SvD infectivity, IgG was purified from CTH522 specific mouse serum pools and from control

pools. A comparative evaluation of IgG and serum showed similar levels of *C.t.* SvD neutralization (Fig. 3a). We continued by demonstrating a correlation between neutralization of *C.t.* SvD and the VD4 region represented by a 17-mer peptide VD4$_{6-22}^{D/E}$: FDTTTLNPTIA-GAGDVK. A strong correlation between the ability to neutralize *C.t.* SvD and VD4$_{6-22}^{D/E}$ peptide specific IgG responses (Two-tailed non-parametric Spearman's rank correlation coefficient, $r$ = 0.86, $p$ < 0.0001) was found (Fig. 3b). Pre-incubation of CTH522 specific sera with CTH522, CTH523, CTH518 (extVD4$^{D/E/F/G}$) and a VD4$_{6-22}^{D/E/F/G}$ construct demonstrated that the VD4 region was solely responsible for neutralization of all four serovars D, E, F, and G. Neutralization was blocked when CTH522 specific mouse serum was incubated with CTH522, CTH518 and VD4$_{6-22}^{D/E/F/G}$, but no inhibition of neutralization

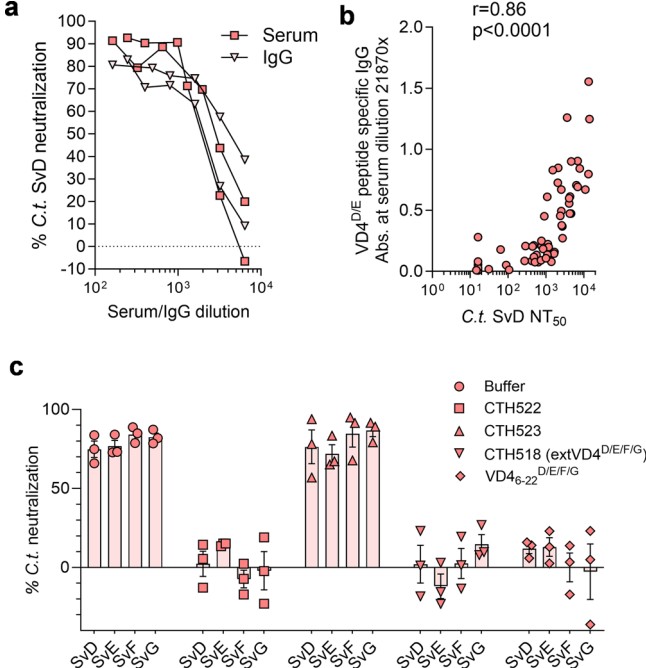

**Fig. 3 | Isotype and specificity of neutralizing antibodies.** IgG was purified from serum pools of CTH522/CAF®01 vaccinated B6C3F1 mice ($n = 100$) and control mice ($n = 100$). **a** Neutralization of *C.t.* SvD using purified IgG were compared to the unfractionated serum samples. Each dot represents the mean neutralization of duplicate readings and results are presented from two individual experiments (serum and IgG from Exp.1 and Exp.2, Fig. 5). **b** Correlation of VD4$_{6\text{-}22}$$^{D/E}$ specific IgG at a serum dilution of 1:21870 ($n = 66$) and the NT$_{50}$ ($n = 66$) was analyzed using Two-tailed non-parametric Spearman's rank correlation coefficient, $r = 0.8641$, 95% CI [0.7838, 0.9160]. **c** CTH522, CTH523, CTH518 (extVD4$^{D/E/F/G}$), and a VD4$_{6\text{-}22}$$^{D/E/F/G}$ construct were incubated with CTH522 specific mouse serum and control serum each diluted 1:200, before being mixed with an equal volume of *C.t.* SvD, E, F, and G and inoculated onto HaK cells. Data points represent the calculated mean NT$_{50}$ of triplicate readings and each bar shows the mean with SEM of three individual determinations. **a–c** Source data are provided as a Source Data file.

was detected with CTH523 comprising the variable domains VD1, VD2, and VD3 (Fig. 3c).

## In vivo effect of CTH522 specific antibodies

To demonstrate a protective role of antibodies in the genital tract, in vivo neutralization was assessed. Mice were infected with *C.t.* SvD precoated with pools of in vitro neutralizing CTH522 specific antibodies from either mice or humans (Fig. 4a). CTH522 specific serum pools as well as control serum pools were incubated with *C.t.* SvD and used directly in our in vitro neutralization assay to confirm neutralization (Fig. 4b) before being inoculated intra-vaginally (i.vag.) into C3H/HeN mice (Fig. 4c). Serum from both mice and humans was diluted either 2 or 4 times with the SvD inoculum. We demonstrated a significant reduced number of inclusion forming units (IFU) in the early phase of infection (Post infection day (PID) 3 and PID7) using both mouse and human CTH522 specific serum pools.

To model neutralization as an antibody effector function in humans, a human immortalized endocervical cell line (End1/E6E7)[36] was used in the in vitro neutralization assay and compared directly to the standard Hamster kidney (HaK) cell line. In contrast to HaK cells, the endocervical cell line is more similar to the epithelial cells lining the endocervix of the female genital tract. Increased infectivity is a possible, but undesirable, outcome of surface-recognizing antibodies due to Fc-receptor mediated enhanced infection. Human serum samples with a SvD NT$_{50}$ > 200 in HaK cells were evaluated using the endocervical cell line. Most serum samples inducing neutralization in HaK

cells also reduced the level of SvD, E, F, and G infection using the endocervical cell line (Fig. 4d).

To further characterize the role of antibodies during the course of a *C.t.* infection we developed a mouse model, which enables *C.t.* SvD to ascend from the lower to the upper genital tract (GT) inducing inflammation and oviduct pathology (Fig. 5a). C3H/HeN mice were depleted of both CD4$^+$ and CD8$^+$ T cells pre and post an i.vag. *C.t.* SvD infection (Supplementary Fig. 6 for depletion status in blood and GT tissue) leading to severe ascending infection, which can be visualized by histochemical staining from day 21 post infection (Supplementary Fig. 7). To characterize the protective role of CTH522 specific IgG in this model, purified IgG from either CTH522 vaccinated or control B6C3F1 mice were injected intraperitoneally (i.p.) into the T cell depleted C3H/HeN mice at PID-3 (and PID15, Exp. 2). The purity and the verification of CTH522 reactivity in the purified IgG fraction are presented in Supplementary Fig. 8. Mice were challenged with $4 \times 10^4$ IFU of *C.t.* SvD and the bacteria levels were measured at several time points post infection in vaginal swabs. Results from two independent experiments are presented in Fig. 5c (Exp. 1: $n = 9$, Exp. 2: $n = 20$–21). We saw a reduction in the overall levels of IFU (Exp. 1, $p = 0.0143$ and Exp. 2, $p = 0.0993$ (ns), Mixed-effect model for repeated measures) due to CTH522 specific antibodies. Ten out of 30 mice (sum of both experiments) receiving CTH522 specific antibodies had a marked effect of the transferred antibodies, with no detectable bacteria at PID7. In contrast, only 1 out of 29 mice controlled the initial infection in the control group. This early control of infection had an impact on the kinetics of bacterial ascension to the oviduct and the progression of pathology. The total levels of *C.t.* SvD were measured in homogenized upper GT tissue (uGT, oviducts and ovaries) of 20 mice (Exp. 1 $n = 4$ and Exp. 2 $n = 16$, sum of both experiments $n = 20$) in each group at PID29 (Fig. 5d). In contrast to the control mice, where half of the mice reached a level of IFU > 1000, none of the mice receiving CTH522 IgG had levels of bacteria above 1000 IFU. Comparing the degree of uGT pathology at PID29 with the IFU levels in the lower genital tract (lGT) showed that mice that either initially controlled the infection and had reduced bacterial levels up to PID14, did not develop uGT pathology within the time period we investigated (Fig. 5e panel 1 + 2, normal oviducts–no pathological changes, no MOMP-positive staining). In contrast, mice that did not control the initial infection and had sustained high levels of IFU in the swabs from PID14 developed substantial oviduct inflammation with dense filling of debris in the lumen and visible EB inclusions (MOMP-positive staining) in both lumen and epithelial cells of the oviducts (Fig. 5e panel 3–5). A FACS analysis of uGT tissue at PID 29 demonstrated the presence of a high percentages of neutrophils especially in mice receiving control antibodies, but also NK cells, macrophages, and dendritic cells (DC's) were present (Supplementary Fig. 9). Even severe pathology with fluid filled hydrosalpinx could be detected at PID29 (Fig. 5e panel 6). These results indicate that CTH522 specific antibodies by reducing the initial load of bacteria can delay the ascension of *C.t.* to the oviduct allowing the T cells in non-depleted mice to enter the genital tract and clear the remaining infection, without the development of neither infection driven nor immune driven inflammation and pathology.

## Protection against *C.t.* ascension

Having demonstrated a possible protective role for antibodies, we next evaluated levels of protection and ascension in an immunocompetent mouse. B6C3F1 mice vaccinated with CTH522/CAF®01 were challenged i.vag. six weeks post last vaccination with $1 \times 10^5$ IFU of *C.t* SvD. Vaccine take (CTH522 specific CMI and Ab responses) and levels of protection determined in lGT, middle (mGT; uterine horns), and uGT tissue were examined (Experimental outline, Fig. 6a). Prior to infection, T cell responses measured as IFN-γ and IL-17A (Fig. 6b) and CTH522 specific antibody responses in both blood (Fig. 6c) and vaginal secretions (Fig. 6d) were determined. A strong T cell response in combination

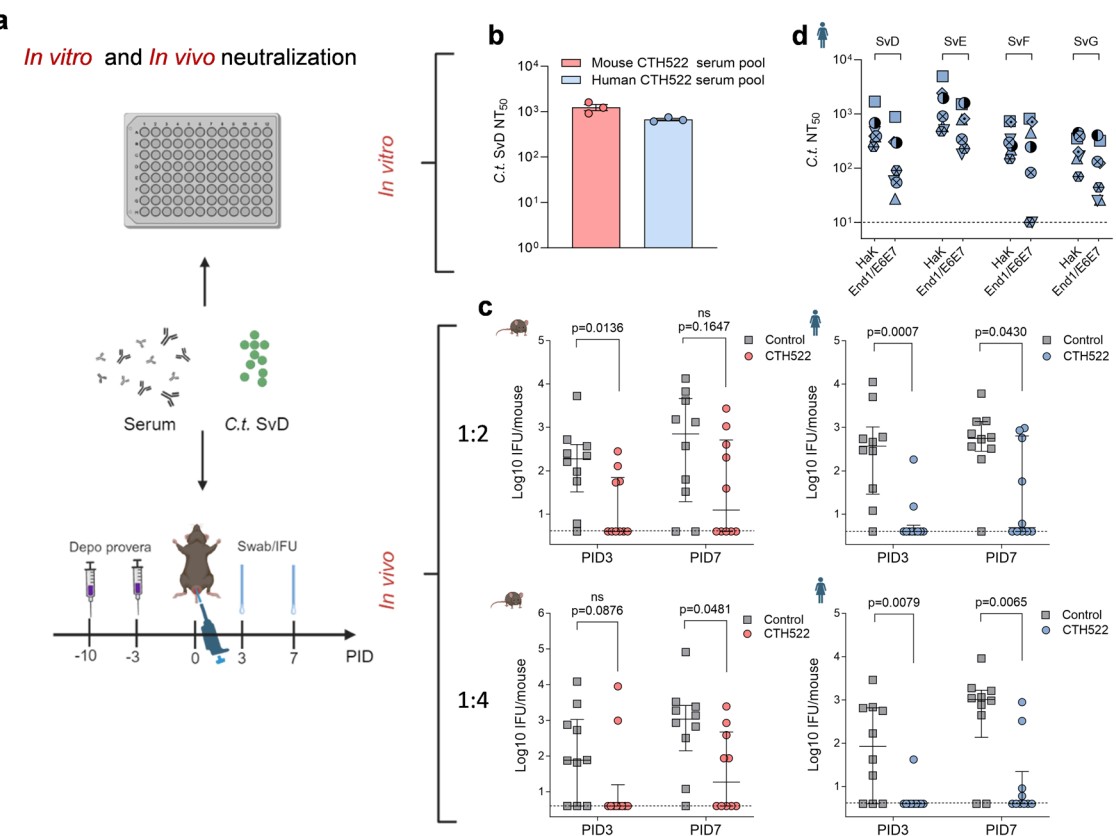

**Fig. 4 | In vitro and in vivo neutralization. a** Experimental overview of in vitro and in vivo neutralization assays. **b** In vitro neutralization: Pools of B6C3F1 mouse and human CTH522 specific serum and control serum were heat inactivated, titrated and mixed with a fixed concentration of *C.t.* SvD, inoculated onto a HaK cell monolayer in duplicates and the $NT_{50}$ were calculated. Each bar represents mean $NT_{50}$ with SEM of three individual determinations. **c** In vivo neutralization. Serum pools were diluted 1:2 or 1:4 with *C.t.* SvD and inoculated i.vag. into depo-provera treated C3H/HeN mice (*n* = 10/group). IFU were determined in swab samples taken at PID3 and 7. Data points represent individual log10 IFU per mouse and line represents median IFU levels with 25th and 75th percentiles. Two-tailed Mann−Whitney *U* test was used for comparison among groups. **d** Comparison of *C.t.* SvD, E, F, and G $NT_{50}$ using human CTH522 specific serum in the standard HaK cell line and a human endocervical cell line (End1/E6E7). Only serum samples (*n* = 7) giving a SvD $NT_{50}$ > 200 in the standard assay were evaluated using the End1/E6E7 cell line. Data points represent $NT_{50}$ calculated from mean of duplicate or triplicate readings. **a** Figure and **c**, **d** Mouse and human symbols were created with Biorender.com. **b**−**d** Source data are provided as a Source Data file.

with high levels of antibodies in serum and significant levels of CTH522 specific IgG in the vaginal secretions were detected. At PID 3, 7, 10, 14, and 17 the vaginal vault was swabbed and levels of IFU determined (Fig. 6e). Significant reduction in IFU was found compared to control mice at all time points. Measuring the area under the curve (AUC) for mice proceeding to PID17 (PID3-PID17) (*n* = 31) demonstrated a highly significant reduction in IFU in CTH522/CAF®01 vaccinated mice compared to control mice (Mann−Whitney *U* test, *p* < 0.0001) (Fig. 6f). Having demonstrated strong protection in the lGT, we continued by investigating the ability of CTH522 induced immune responses to protect against the infection in the mGT and uGT tissue. In immunocompetent mice *C.t.* reaches the oviducts less frequently compared to T cell depleted mice. A total of three independent experiments were performed. Mice were challenged i.vag. with doses ranging from $4 \times 10^4$–$1 \times 10^5$ IFU and at PID21-22 mGT and uGT were isolated and homogenized and IFU determined (Exp. 1 *n* = 16, Exp. 2 *n* = 12, Exp. 3 *n* = 8). Results from the individual experiments and a sum of the three experiments are presented in Fig. 6g. In CTH522/CAF®01 vaccinated mice a total of 2 out of 36 mGT samples were *C.t.* positive. For comparison, half of the mGT samples from the control mice (18 out of 36) were positive for *C.t.* (*p* < 0.0001, two-sided Fisher's exact test). Similar results were found in the homogenates of uGT where 1 out of 36 samples were positive in CTH522 vaccinated compared to 13 out of 36 in the control group (*p* = 0.0006, two-sided Fisher's exact test). These results demonstrated strong protection

against upper genital tract infection in CTH522/CAF®01 vaccinated mice.

## Longevity of immunity and protection

Induction of long-lived immunity is essential for a Chlamydia vaccine since the vaccine will likely be introduced in early teens before sexual debut. Using CAF®01 as an adjuvant for Tuberculosis vaccine candidates, we have previously demonstrated induction of long-lived immunity both in mice[37,38] and humans[39]. To examine the longevity, quality, and efficacy mice were rested for a period of up to 1 year post last CTH522/CAF®01 vaccination and the immune responses were assessed at various time points (Fig. 7a). The antibody responses were measured 8, 31, and 56 weeks post 1st vaccination and during the 1 year resting period the CTH522 specific IgG titer in the blood decreased 67-fold (AEU:78753/1169 = 67) (Fig. 7b). In parallel with the decrease in serum antibody levels, the neutralizing capacity fell to a median $NT_{50}$ of around 100 in week 31 and to the detection limit of the neutralization assay at 1 year (Fig. 7b). The vaccine specific CMI responses were assessed by measuring the IFN-γ recall response in in vitro CTH522 stimulated splenocytes at week 6 and week 56 post 1st vaccination. The response remained remarkably stable with a median IFN-γ release of around 40,000 pg/ml at both time points (Fig. 7b). By intracellular cytokine staining (ICS) staining for IFN-γ, IL-2, TNF-α, and IL-17A, CTH522 specific CD4+ T cells were subsequently characterized. We found a median level of 1.02% cytokine producing CD4+ T cells in

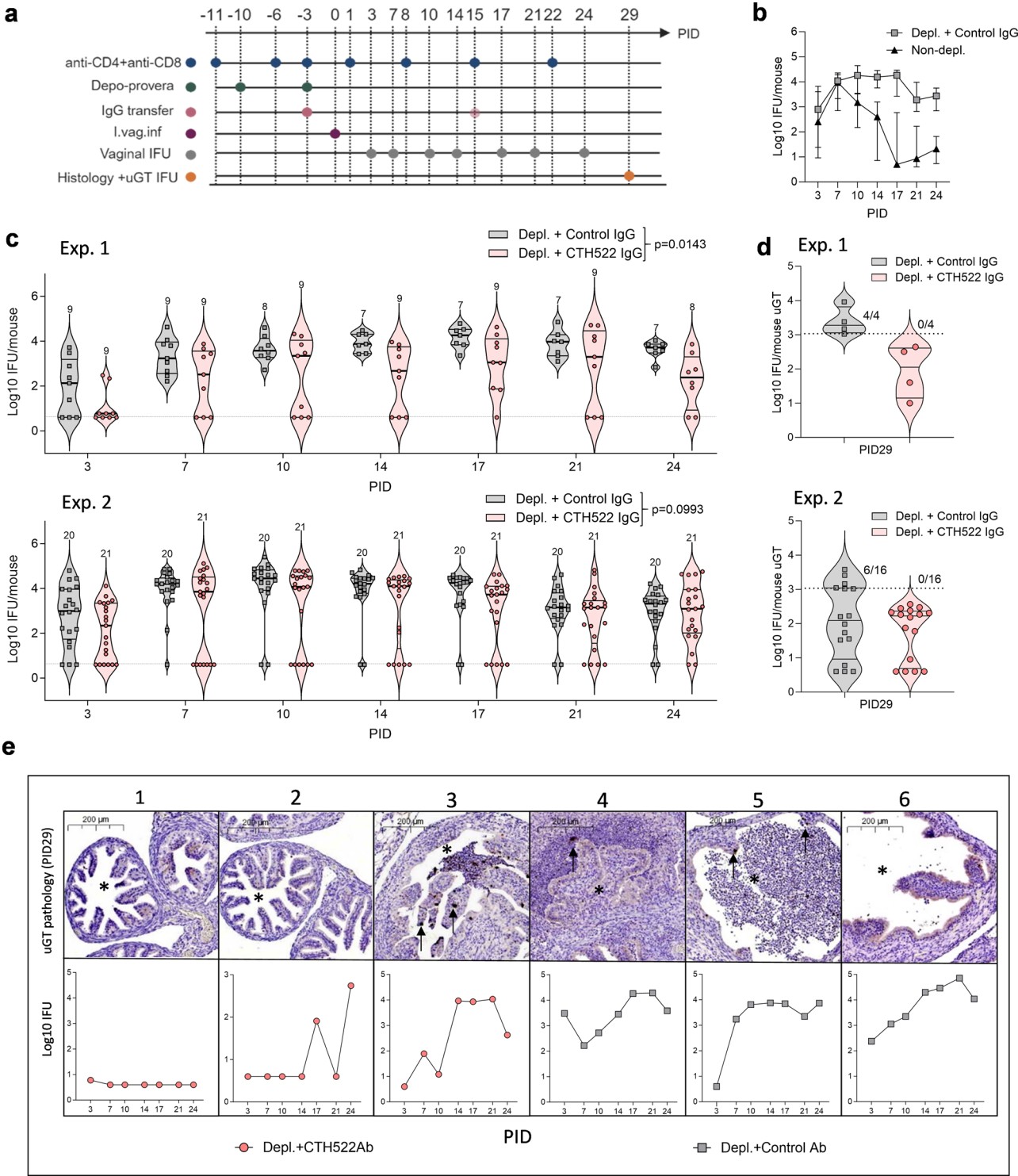

**Fig. 5 | The role of CTH522 specific antibodies during *C.t.* infection. a** Overview of the experimental outline and treatments. Groups of C3H/HeN mice were CD4+/CD8+ depleted before and during an i.vag. *C.t.* SvD infection. A non-depleted group was included as control. At PID-3, depleted mice received either purified IgG from control or CTH522 vaccinated B6C3F1 mice. In Exp. 2 mice received an additional dose at PID15. **b** Kinetics of clearance in CD4+/CD8+-depleted mice receiving IgG from control or CTH522 vaccinated mice (PID 3 and 7 *n* = 29, PID10 *n* = 28, PID14-24 *n* = 27) compared to non-depleted mice (*n* = 30). Levels of IFU were determined in vaginal swab samples at several time points post infection and compared to levels obtained in non-depleted mice. Results are presented as median Log10 IFU/mouse with 25th and 75th percentiles. **c** Kinetics of *C.t.* SvD clearance of CD4+/CD8+ depleted mice receiving IgG purified from either CTH522 vaccinated or control mice. Two individual experiments (Exp. 1 and Exp. 2) are presented. Results are presented as

Log10 IFU/mouse and presented as a violin plot with median and 25th and 75th percentiles, all data points. The number of mice in each group are shown in the violin plot. Mixed-effect model with the Geisser-Greenhouse correction for repeated measures were used for comparison among groups. **d** Levels of IFU in homogenates of uGT (oviducts/ovaries) at PID29 in the two individual experiments (Exp. 1 *n* = 4, Exp. 2 *n* = 16). The data are presented as a violin plot plots with median and 25th and 75th percentiles, all data points. Dotted line indicates the 1000 IFU level **e** Histological sections of 6 oviducts (5x magnification) from mice with different *C.t.* clearance kinetics (Panel 1–6) stained with an in-house rabbit anti-MOMP antibody visualizing the *C.t.* inclusions. The oviduct lumens are denoted with asterisks and *C.t.* inclusions with arrows. The presented oviduct is representative for the mouse examined and only performed on a total of 7 mice in Exp. 1. **b**–**e** Source data are provided as a Source Data file.

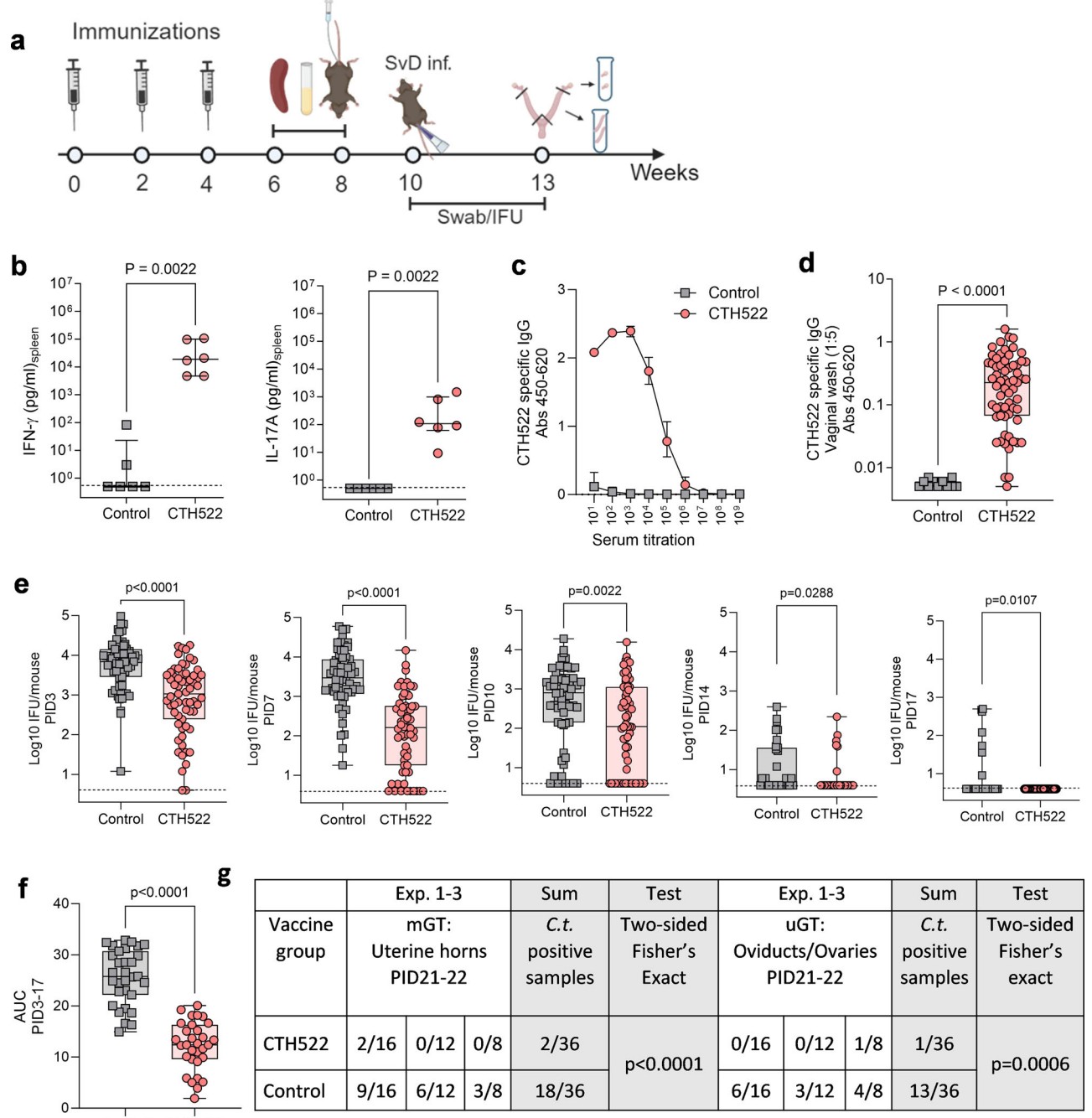

**Fig. 6 | Protective effect of CTH522/CAF®01 induced immune responses.**
**a** Experimental outline. B6C3F1 mice were immunized with 10 µg CTH522/CAF®01 or CAF®01 alone. **b** CTH522 specific T cell responses. Splenocytes ($n = 6$) were stimulated in vitro with CTH522 and IFN-γ and IL-17A responses were measured in supernatants. Data points represent cytokine levels in antigen stimulated wells after subtraction of media only for each mouse and each line represents median level with 25th and 75th percentiles. Two-tailed Mann−Whitney $U$ test was used for comparison among groups. **c** Serum from vaccinated mice ($n = 63$) and control serum samples ($n = 6$) were isolated 4 weeks post last vaccination and CTH522 specific IgG analyzed by ELISA. Data points represent the median OD value with 25th and 75th percentiles at each titration step. **d** Vaginal wash samples were collected from CTH522 vaccinated mice ($n = 63$) and control mice ($n = 11$), diluted 5 times, and CTH522 specific IgG were measured. Results are presented as a Box and Whiskers plot with median and 25th and 75th percentiles (Box) and Min to Max, all data points (Whiskers). Two-tailed Mann−Whitney $U$ test was used for comparison among groups. **e** Mice were challenged i.vag. with $1 \times 10^5$ IFU/mouse of *C.t.* SvD. IFU

were recovered from vaginal swabs at day 3, 7, 10 ($n = 63$), 14, and 17 ($n = 31$) post infection. Data are presented as a Box and Whiskers plots with median and 25th and 75th percentiles (Box) and Min to Max, all data points (Whiskers). Two-tailed Mann−Whitney $U$ test was used for comparison among groups. **f** Area under the curve (AUC) from PID3 to PID17 was calculated for the 31 mice proceeding to PID17 and results presented as a Box and Whiskers plot with median and 25th and 75th percentiles (Box) and Min to Max, all data points (Whiskers). Two tailed Mann−Whitney $U$ test was used for comparison among groups. The experiment has been repeated three times with similar results. **g** Protection against ascending infection divided into mGT; Uterine horns and uGT; Oviducts/ovaries. The numbers of culture positive samples/total number in the individual experiment as well as a sum of the three experiments are presented. Sum of mGT *C.t.* positive vs. negative in the two groups, Two-sided Fisher's exact test, $p < 0.0001$. Sum of uGT *C.t.* positive vs. negative in the two groups, Two-sided Fisher's exact test, $p = 0.0006$. **a** Figure was created with Biorender.com. **b**–**f** Source data are provided as a Source Data file.

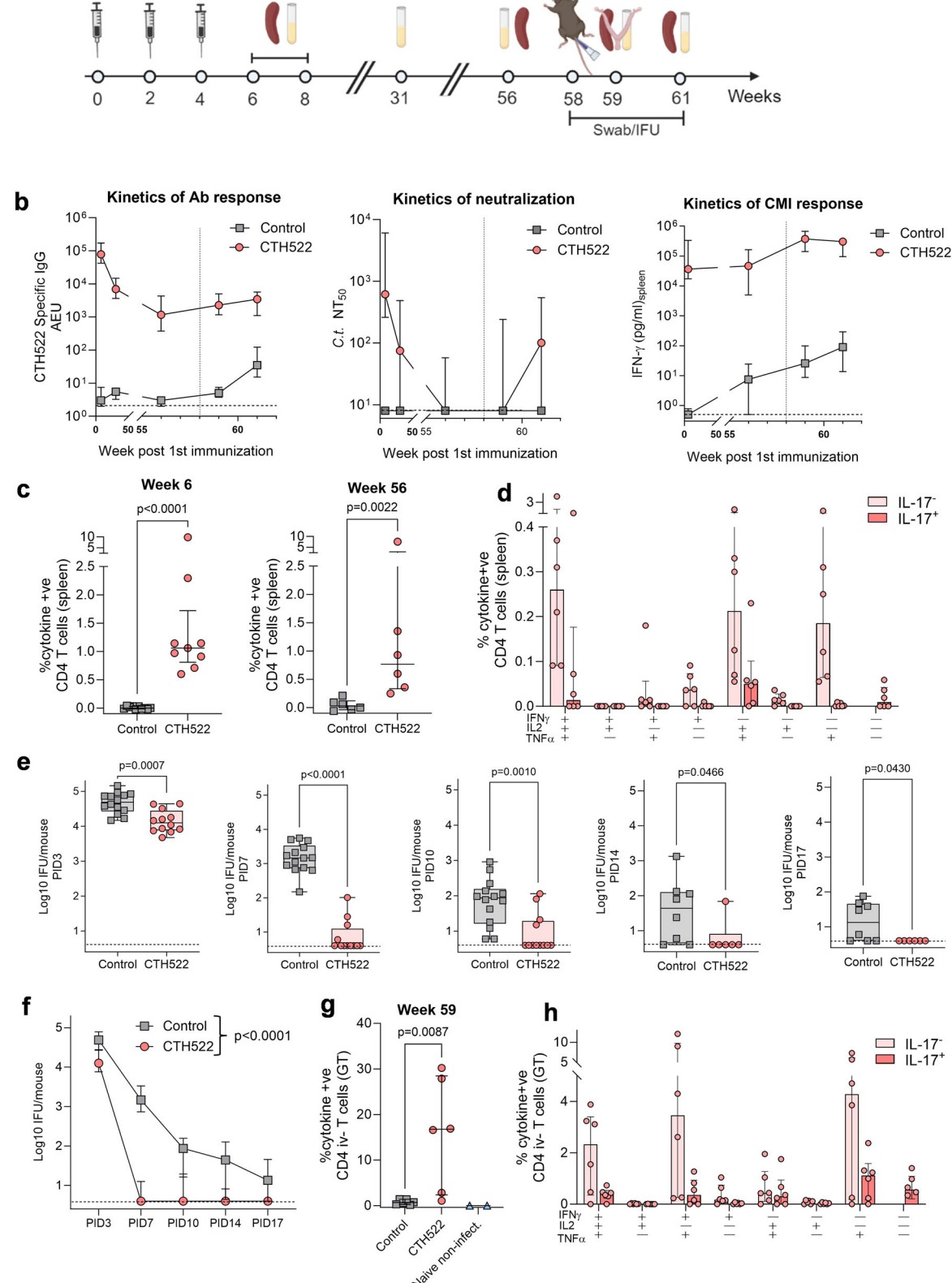

the spleen 6 weeks post vaccination and this level was only slightly reduced to 0.77%, 56 weeks post 1st vaccination (Fig. 7c). The CD4$^+$CD44$^{high}$ T cell populations were characterized based on their ability to secrete the effector cytokines IFN-γ, TNF-α, IL-2, and IL-17A in any combination and separated into IL-17A$^-$ and IL-17A$^+$ T cells. Multi-functional CD4$^+$ T cells expressing TNF-α$^+$ and IL-2$^+$, triple-positive cells

expressing IFN-γ$^+$IL-2$^+$TNF-α$^+$ or IL-17A$^+$ IL-2$^+$TNF-α$^+$, and TNF-α single positive cells dominated the responding T cell population in the spleen 56 weeks post 1st vaccination, representing both Th1 and Th17 effector memory and central memory T cells (Fig. 7d). At week 58 post vacci-nation, the mice were challenged i.vag. with $1 \times 10^5$ IFU of *C.t.* SvD and the ability of the vaccine to control the infection was determined in

**Fig. 7 | Longevity of CTH522 specific immunity and protection. a** Experimental outline. B6C3F1 mice were immunized with 10 µg CTH522/CAF®01 or Sham-immunized. **b** Kinetics of CTH522 specific IgG: week 8, 31, 56 n = 20 (except CTH522 week 56 n = 18), week 59 and 61, n = 6, NT_50: CTH522 week 8, 31, 56 n = 19 (except week 56 n = 17), Controls n = 20, week 59 and 61, n = 6 and IFN-γ responses: n = 6 (except Controls week 61 n = 4). Dotted vertical line indicates time of challenge. **c** Six (n = 9) and 56 (n = 6) weeks post 1st vaccination, splenocytes were stimulated with CTH522. Data points represent the percentages of cytokine positive (IFN-γ, IL-2, IL-17A, TNF-α) CD44^high CD4^+ T cells out of all CD4^+ T cells. **d** Data points (n = 6) represent the percentage of IL-17A^- (Th1) and IL-17A^+ (Th17) out of all CD4^+ T cells producing the cytokine TNF-α, IL-2, and IFN-γ in any combination. **e, f** 58 weeks post 1st vaccination mice were challenged i.vag. with *C.t.* SvD and IFU recovered from swabs at PID3, 7, 10 (Sham-immunized n = 14, CTH522 immunized n = 12), 14 and 17 (Sham-immunized n = 8, CTH522 immunized n = 6). **e** Data points

represent Log10 IFU/mouse and are presented as Box and Whiskers plots with median log10 IFU and 25th and 75th percentiles (Box) and Min to Max, all data points (Whiskers). **f** Kinetics of clearance, median log10 IFU/mouse at PID3-17 for CTH522/CAF®01 and Sham-immunized mice. **g** At PID10 (week 59) GT tissue were stimulated with CTH522 (n = 6). Non-infected mice were included as controls (n = 2). The percentages of cytokine positive (expression of any of the cytokines IFN-γ, IL-2, IL-17A, TNF-α) CD44^high CD4^+ iv- GT T cells out of all CD4^+ iv- T cells were analyzed by flow cytometry. **h** Data points represent the percentage of IL-17A^- (Th1) and IL-17A^+ (Th17) out of CD4 iv- GT T cells producing the cytokine TNF-α, IL-2, and IFN-γ in any combination. **b–d, f–h** Each data point, bar, or line represents median levels with 25th and 75th percentiles. **c, e, g** Two-tailed Mann–Whitney U test. **f** Mixed-effect model for repeated measures. **a** Created with Biorender.com. **b–h** Source data are provided as a Source Data file.

vaginal swabs at PID3-17. Significant protection was detected at all time-points (Fig. 7e). The most pronounced reductions in IFU were detected from PID3 to PID7 where we found a 3147-fold reduction in median IFU levels (12589/4) in the CTH522/CAF®01 vaccinated mice compared to 33-fold reduction in median IFU levels (49091/1469) in the control group. The strong level of protection detected 1-year post vaccination was similar to previous finding after a prime boost vaccination strategy with two different doses of CTH522 (Supplementary Fig. 10). The CTH522 specific antibody and CMI responses were evaluated post infection (PID10 and PID21). After infection, a primary CTH522 specific IgG response developed 3 weeks post infection in the control group and we detected a boost in CTH522 specific neutralizing IgG in the serum of vaccinated mice (Fig. 7b). The same overall results were found regarding the CTH522 specific CMI response. Here, we also saw the development of a primary CTH522 specific IFN-γ response in the control group and a boost of the response in CTH522/CAF®01 vaccinated mice (Fig. 7b). A characterization of the CTH522 specific T cells in the GT 10 days after challenge demonstrated a median level of 15% cytokine positive CD4^+ T cells (Fig. 7g) composed of substantial Th1 (IFN-γ^+ TNF-α^+ double positive and TNF-α^+ single positive cells) and Th17 (IL-17A^+ TNF-α^+ double positive, IL17^+ single positive cells) T cells (Fig. 7h) In contrast, in non-vaccinated infected animals, we only detected very low levels of CTH522 specific T cells. Gating strategies for flow cytometry analyses are shown in Supplementary Fig. 11. Thus, overall we found long-lasting immunity and protection, including vaccine re-call responses of both CMI and antibodies one year after vaccination.

## Discussion

Chlamydia vaccine development has recently advanced with the first ever Chlamydia vaccine to reach clinical trial. The vaccine CTH522/CAF®01 completed phase I clinical trial successfully and was safe and immunogenic[2]. CTH522, a molecular engineered recombinant version of MOMP, was designed to capture B and T cell epitopes in a molecule suitable for GMP production and clinical development[10,29]. In the current study we describe how protective immune signatures from the mouse model translate to humans.

We previously showed that CTH522/CAF®01 induced a robust human cellular response, measured by the numbers of vaccine-specific IFN-γ secreting T cells by ELISpot[2]. Here, we further characterized the human T cell response and showed that in vitro CTH522 stimulated PBMCs led to secretion of both IL-17A, the hallmark of Th17 cells, as well as IFN-γ, IL-6, IL-13, and TNF-α as dominating cytokines, a profile that resembles the cytokine profile of CAF®01 in mice[31]. Cellular immunity, specifically T cells secreting IFN-γ are pivotal in Chlamydia immunity[6]. Th17 cells display a great degree of plasticity and although they can be pathogenic[40], Th17 cells are important for the control of mucosal pathogens[41]. Evidence support that Th17 cells have similar roles in humans and mice[42] and studies in mice show that Th17 cells have the ability to home to mucosal tissues, recruit neutrophils,

antigen presenting cells and facilitate IgA secretion[41,43]. A recent study by ref. 44. highlighted that a T cell signature of central memory CD4 Th1 and Th17 cells are associated with reduced *C.t.* reinfection in a highly exposed cohort. Until now no commercially available vaccine adjuvant has been shown to induce a robust Th17 response in humans, which makes the Th17 response induced by CAF®01 in mice and humans a unique feature for this adjuvant system. A limitation to the current study was the low number of human PBMC samples for a more detailed characterization of T cell phenotypes.

As serovar coverage is central for a future Chlamydia vaccine, CTH522 incorporates sequences from *C.t.* serovars D-G (CTH518, Fig. 1a) on a backbone of the SvD sequence of MOMP (CTH523, Fig. 1a), thereby benefitting from both the serovar neutralizing epitopes and stretches of conserved T-cell epitopes within MOMP[25]. Detailed T cell epitope mapping with overlapping peptides showed that human responses were in general broader and more widely distributed throughout the MOMP sequence compared to the inbred mouse responses, which were focused on fewer epitopes. The peptides P10^D and P19-20^D strongly recognized by humans are localized in the highly conserved segments CSII and CSIII, respectively, that are ≥95% conserved between genital *C.t.* serovars (Supplementary Table 1) and corresponds to previously published human T cell epitope regions within MOMP found during a natural infection[32,33].

Antibody responses against CTH522 are characterized by the mouse IgG subclasses IgG1, IgG2a/c and IgG2b and human subclasses IgG1 and IgG3 (Fig. 2c, d). This subclass profile resembles findings from naturally infected individuals[45], and fits well with immunity to intracellular pathogens. Serum antibodies from vaccinated mice and humans could neutralize *C.t.* SvD, E, F, and G infection in vitro, using the HaK cell line[46] (Fig. 2f,g). Studies in HaK cells measure the effect of antibodies in a non-Fc-dependent manner[47], and since *C.t.* uptake by epithelial cells can be influenced by surface expression of FcγR or FcRn[48], we also tested human serum for neutralizing capacity in immortalized human endocervical epithelial cells End1/E6E7[36] and confirmed the neutralizing ability of the antibodies and no enhanced infection was detected (Fig. 4d).

Mapping of CTH522 specific B cell epitope demonstrated a great degree of similarity across species and these epitopes were localized to surface exposed variable domain regions of MOMP in both mice and humans. However, in contrast to mice, a strong reactivity against P21-22^D in close proximity to the VD3 region was identified in humans. A predominant recognition of surface exposed regions of CTH522 was also observed in non-human primates[49] and minipigs[50]. Blocking of *C.t* SvD, E, F, and G neutralization with a construct representing the VD4 region for all serovars D-G (Fig. 3c) demonstrated that the neutralizing epitopes in CTH522 were located in the VD4 region. We further demonstrated that CTH522 specific antibodies were able to neutralize serovars present in the B (SvD, E, B), B-related (SvF and SvG), and C-related complex (SvK) and to a lesser extent the C complex serovars (SvH, I, Ia, J, C), except for SvA where we did see detectable levels of

neutralization. This serovar restricted neutralization likely reflects to which degree the VD4 region is surface accessible in the individual serovars[34]. CTH522 will therefore target the majorities of circulating strains with neutralizing and surface recognizing antibodies in combination with a broad and cross-protective CMI response which potentially can cover all *C.t.* serovars. Recently, we published a paper focusing on VD1 based vaccine constructs from SvA and SvJ demonstrating VD1 based neutralization of C and C-related complex serovars[35]. In the present study, we likewise demonstrated strong neutralization of C complex serovar (except for SvH) with an extVD1 immuno-repeat based on SvIa (extVD1[Ia]*4) (Supplementary Fig. 5). Our previous results on VD1 based vaccine construct and the present study agrees with what has previously been reported on MOMP specific monoclonal antibodies[51–54]. As with the HPV vaccine, it is therefore a possible scenario that the CTH522 vaccine can be further improved along the way, when its efficacy in humans has been established (Phase IIb/III). The vaccine could e.g., be supplemented with an immuno-repeat based on VD1 regions from C complex serovars in order to further broaden the serovar coverage.

CTH522 specific neutralizing serum from both humans and mice significantly protected mice when inoculated in a mixture with *C.t.* SvD bacteria i.vag. into C3H/HeN mice (Fig. 4c). Since human IgG bind to mouse FcγR with similar binding strengths as to human ortholog receptors[55] our results suggest that antibody mediated control of infection could be neutralization and/or other Fc mediated antibody effector functions. Even though bacteria were coated with immune sera, there were still mice that failed to control the infection. Likewise, adoptively transferred CTH522 specific IgG failed to reduce the initial level of bacteria in the lGT of 20 out of 30 mice. Within the investigated time period (29 days) we did, however, see an overall lower burden in the uGT in the group receiving CTH522 antibodies compared to controls. This highlights that controlling chlamydial infections and targeting the bacteria's biphasic life cycle, which has an intracellular and extracellular stage, requires an adaptive T cell response in addition to antibodies. This is also in line with our previous findings, where adoptive transfer of vaccine-specific antibodies to Rag1 KO mice completely protected half of the mice[26].

In the present study, we investigated the protective role of CTH522 induced antibodies in regard to the kinetics of *C.t.* ascension from lGT to uGT in a T cell depleted, *C.t.* susceptible, C3H/HeN mouse strain[56]. It has been suggested that innate immune responses alone spontaneously resolve genital infection with *C.t.* in contrast to what is seen for *C. muridarum* in mice[57]. However, in the present study we demonstrated that depleting mice of T cells led to ascension of *C.t.* SvD and development of uGT pathology, which could be detected at PID29. We further observed that adoptively transferred CTH522 specific IgG could change the kinetics of ascension to the uGT in a number of mice, delaying the arrival of bacteria to the oviducts and the progression of pathology (Fig. 5e). A similar finding was noted by ref. 8, using the "backpack" tumor system to deliver MOMP specific MAbs into the serum and vaginal secretions of mice. In a non-human primate experiment with a native MOMP subunit vaccine, reduction in ocular shedding was observed, albeit no difference from control animals in progression of ocular disease[9]. This is in line with our observations where we detected an early decrease in burden in the lGT which was diminished over time due to lack of T cells (Fig. 5c). Antibodies can initially control infection by neutralizing *C.t.* and it is well known that antibodies can be efficiently transferred to the female GT, either by transudation, reviewed in ref. 58, or by the MHC class I-related neonatal Fc receptor[59,60], a mechanism by which antibodies have been shown to protect against vaginal infection with HSV-2 and HIV-1[60,61]. Once antibodies have reached the vaginal mucus, they can accumulate on the surface of the bacteria and either directly inhibit them from binding the epithelial target cells, activate the

complement system leading to direct lysis of the bacterial membrane, or facilitate opsonization and/or ADCC[11].

Despite the fact that CTH522 antibodies alone, either by directly coating the bacteria (in vivo neutralization, Fig. 4c) or when adoptively transferred to T cell depleted mice (Fig. 5c,e), was not sufficient to completely control the infection in all mice, the combined/synergistic response of antibodies and CMI in immunocompetent mice was highly protective. We show that vaccination with CTH522/CAF®01 in immunocompetent mice significantly protect against infection in the lGT (Fig. 6e) and against the ascending bacteria in the mGT and uGT (Fig. 6g). These mice contained both neutralizing antibodies in the GT (Fig. 6d) as well as cellular immunity (Figs. 6b and 7c). Similarly, to mice and humans, Lorenzen et al.[49] found that CTH522/CAF®01 vaccinated Cynomolgus macaques established a strong and neutralizing antibody response in combination with a multifunctional CD4+ T cell response. The vaccine did not significantly accelerate clearance of infection compared to the control animals, possibly due to the very high challenge dose of $5 \times 10^7$ IFU of *C.t.* SvD/animal needed to establish infection in the NHP model, a dose that is far from the 675–15,920 copies/ml found in semen of infected patients[62]. We speculate, that this high dose model may call for an immune response encompassing all arms of the immune system, including CD8+ and CD4+ T cells, as well as neutralizing antibodies, in order to reduce the bacterial level.

Since a future Chlamydia vaccine is likely to be introduced in parallel with the HPV vaccine in adolescents before sexual debut, immunity needs to be sustained for at least a decade. To model this, we used the mouse to show that vaccine induced antibody and Th1/Th17 responses were long-lived up to 1-year post vaccination (Fig. 7b, c). Although a mouse lifespan cannot be directly translated to humans, one mouse year approximately corresponds to several decades in human years, from teens to mature and middle age[63]. The long-lived T cell response consisted of multifunctional CD4+ T cells being either triple positive (TNF-α/IL-2/IFN-γ and TNF-α/IL-2/IL-17A), double positive (TNFα/IL-2) or single positive (TNF-α) expressing T cells. Multiple studies have demonstrated that the most effective memory T cells are cells producing high levels of multiple cytokines (Poly or multifunctional T cells)[37,38,64]. Especially, less differentiated IL-2 producing cells are superior and less likely to undergo apoptosis compared to highly differentiated effector cytokine producing T cells[65].

Serum antibody levels decreased 11-fold after 6 month and 67-fold during the one-year period. This antibody trajectory likely means that over time the role of neutralizing antibodies in the immediate control of infection will decline, and instead rely on the memory T and B cell response (Fig. 7c). In fact, we measured a significant reduction in IFU levels already at PID3, which became more pronounced at PID7 where the level of bacteria in the CTH522/CAF01 group was reduced from a median log10 IFU of 4.1 (PID3) to 0.6 (PID7). We saw a recall antibody and CMI response following infection, slightly offset bacterial clearance, which could impact clearance of the infection and bacterial shedding. In a recent study by ref. 66 a *C. muridarum* MOMP vaccine formulated in Montanide and Montanide/CpG, administered by different routes, protected mice up to day 180[66]. CTH522 long-term immunity consisted of multiple effector functions, and we saw fast recruitment of Th1/Th17 cells into the genital tract, as well as a vaccine recall CMI and IgG response in spleen and blood (Fig. 7). Importantly, the stable immunological CMI profile for CAF®01 has been confirmed in a Tuberculosis vaccine phase I trial, showing immune responses for more than 150 weeks[39].

A limitation of the current study is the use of human *C.t.* serovars in the mouse model, as this does not fully reflect the pathogenesis and immune responses of human chlamydial infections. Especially, IFN-γ-mediated effector mechanisms leave *C.t.* more vulnerable in the mouse model, which means both T cell and antibody effector mechanisms could be overestimated[67]. Another limitation is to what extent the

in vivo neutralization assay mimics the antibody interaction with *C.t.* in cervicovaginal mucus. We preincubated for 30–45 min. with a high level of antibody-containing serum, as done by ref. 68, which informs to what extent we can block *C.t.* from infecting epithelial cells. Although antibody levels will be significantly lower in the GT following vaccination, it has been shown that antibodies, even in low concentrations, interact with the cervical mucin and entrap bacteria and viruses in the mucin mesh, which could slow down the interaction with the epithelial cells, as seen during HSV infection[69].

To conclude, CTH522/CAF®01 induces a shared Th1/Th17 T cell profile in mice and humans, as well as functional antibodies with both in vitro and in vivo neutralizing activity. The vaccine has broad serovar coverage via conserved T cell epitopes and broadly *C.t.* recognizing antibodies. The vaccine induces sustained immunity which is key since the end-game for Chlamydia vaccine development is a vaccine given to young teenagers before their sexual debut[70]. Our study will significantly advance translational research and clinical development of chlamydia vaccines and pave the way for further development into a phase IIb study.

# Methods

## Animals
Female B6C3F1/OlaHsd (H-2b,k)(C57BL/6JOlaHsd inbred female x C3H/HeNHsd inbred male) and C3H/HeNHsd (H-2k) mice, 6–8 weeks of age, were obtained from ENVIGO, The Netherlands. The mice were housed under standard environmental conditions and provided standard food and water *ad libitum*. Before the initiation of experiments, mice had at least one week of acclimatization in the animal facility. Mice had access to irradiated Teklad Global 16% protein Rodent Diet (Envigo, 2916c) and water ad libitum. Mice were housed at an ambient temperature of 20–23 °C and 45–65% relative humidity on a 12 h/12 h light/dark cycle with 15 min dusk and dawn transition periods under Biosafety Level (BSL) II conditions in individually type III ventilated cages (Scanbur, Denmark) and had access to nesting material (Enviro-Dri) as well as enrichment (aspen bricks, paper house, corn, seed, and nuts, Brogaarden).

Animal experiments were conducted in accordance with regulations of the Danish Ministry of Justice and animal protection committees by Danish Animal Experiments Inspectorate Permit 2018-15-0201-01502 and in compliance with European Union Directive 2010/63/EU. The experiments were approved by a local animal protection committee at Statens Serum Institut, IACUC, headed by DVM Kristin Engelhart Illigen.

## Samples from clinical trial participants
Samples from healthy females aged 18–45 years were taken from our first in human clinical trial study[2]. The study was done in accordance with the International Conference on Harmonization's Good Clinical Practices guidelines, and is registered with ClinicalTrials.gov, number NCT02787109. The study protocol was approved by the London–Chelsea Research Ethics Committee, the Research and Development department at Imperial College Healthcare National Health Service (NHS) Trust, and the Medicines and Healthcare Products Regulatory Agency (EudraCT number 2015-004330-10). All participants gave written informed consent before enrollment. The analyzed human serum samples were collected at baseline and at day 126 post 1st immunization (two weeks post 3rd intramuscular (i.m.) immunization with 85 μg CTH522/CAF®01). The analyzed human PBMC samples were collected at day 140 post 1st immunization (two weeks post 3 x i.m. (3 × 85 μg CTH522/CAF®01) + 1 x i.n. (60 μg CTH522)). PBMC samples from saline immunized humans were used as controls.

## Cultivation and harvesting of *C.t*
*C.t.* SvD/UW-3/Cx, *C.t.* SvE/Bour, *C.t.* SvF/IC-Cal-3, *C.t.* SvG/UW-57/Cx, SvH/UW-43/Cx, SvI/UW-12/Ur, SvJ/UW-36/Cx, SvK/UW-31,SvA/HAR-13,

SvC/TW-3 (all from ATCC), SvIa/sotonIa3/Ia870 (from the Chlamydia Biobank), and SvA/2497 and SvB/Tunis-864 (from LSHTM) were propagated in HeLa-229 cells (ATCC CCL-2.1™) or McCoy cells (ATCC CRL-1696™) for 2–3 days and harvested by repeated centrifugation and sonication steps. Finally, the bacterial suspension was layered on a 30% renografin solution and centrifuged at 40,000 g for 30 min. After centrifugation, the pellet was re-suspended in a sucrose-phosphate-glutamate (SPG) buffer and stored at −80 °C. Serovar typing of the bacteria was confirmed by chromosomal DNA extraction, PCR amplification and sequencing of the gene and flanking regions of *ompA*. All *C.t.* serovars were tested negative for mycoplasma (Mycoplasma laboratory, SSI). The concentration (IFU) of the *C.t.* SvD batch was quantified by titration in McCoy cells. Protein concentrations were determined by bicinchoninic acid protein assay (BCA) (Pierce, Thermo Fisher Sci., Waltham, Massachusetts, US).

## Antigen cloning and purification
The CTH522 batch was of good manufacturing practice quality produced at Statens Serum Institut[22,29]. CTH518 (identical to Hirep2)[10] and CTH523 constructs were based on the amino acid sequences shown in Fig. 1a with addition of six N-terminal histidines. Synthetic DNA constructs were codon-optimized for expression in *E. coli* followed by insertion into the pJexpress 411 vector (ATUM, Newark, CA, USA). Likewise, VD4$_{6-22}$$^{D/E/F/G}$ is a fusion protein with an N-terminal six histidine tag holding 17 amino acids from VD4 of the serovars D (FDTTTLNPTIAGAGDVK), E (FDTTTLNPTIAGAGDVK), F (VDITTLNPTIAGSGSVA), and G (VDITTLNPTIAGSGSVV). In all constructs cysteines were exchanged with serines to avoid disulphide bridge formation during recombinant production. Purification was done by induction of expression with IPTG in *E. coli* BL-21 (DE3) cells transformed with the synthetic DNA constructs. Inclusion bodies were isolated and extracts were loaded on a HisTrap column (GE Healthcare, Chicago, Illinois, USA), followed by anion exchange chromatography on a HiTrap Q HP column and dialysis to a 20 mM glycine buffer, pH 9.2. Protein concentrations were determined by BCA assay.

## Synthetic peptides
PepSets of 20–21-mer peptides with 10aa overlap covering CTH522 were produced by GeneCust (Boynes, France) (Supplementary Table 1).

## Immunization
B6C3F1 mice received a total of three immunizations at two-week intervals subcutaneously (s.c.) at the base of the tail in a total volume of 200 μl. The vaccines consisted of 10–25 ug of antigen diluted in Tris-buffer with 2% glycerol (pH 7.4) and mixed by vortexing with adjuvant consisting of 50 μg/dose of the glycolipid trehalose 6,6'-dibehenate (TDB) incorporated into 250 μg/dose of cationic liposomes composed of dimethyldioctadecylammonium (DDA) (CAF®01). In Exp. 2 (Fig. 5) mice, in addition, received a single dose of 25 μg CTH522 intranasally without adjuvant. Sham-immunized mice were included as controls.

## Mouse and human cell preparation
Splenocytes were isolated from individual mice at week 6, 56, 59 (PID10) and 61 (PID21) and GT tissue was isolated at week 59 (PID10). Samples were obtained from 6–9 mice per group in RPMI 1640 (Thermo Fisher Sci, Gibco. cat. #21875-034). 250 μl of anti-CD45.2 FITC (BD Pharmingen, clone 104, cat. #560695, 2.5 μg) was intravenously (iv) injected via the tail vein of each mouse 3–6 min prior to organ harvest to distinguished genital tract-localized (iv-) and vasculature-associated (iv +) cells. The organs were homogenized through a 100 μm nylon filter (Falcon). In addition, genital tracts (GTs) were incubated before homogenization for 1 h at 37 °C, 5% CO$_2$ in type IV collagenase (0.8 mg/ml) (Sigma) and 30 min in DNAse I (Roche) (0.08 mg/ml) and processed both before and after incubation with

gentleMACS™ Dissociator (Miltenyi Biotec). Washed and centrifuged (700 × *g*, 5 min.) cell pellets from all organs were resuspended in RPMI-1640 (Thermo Fisher Sci., Gibco, cat. #21875-034) supplemented with 1% (vol/vol) L-glutamine, 1% non-essential amino acids, 1% sodium pyruvate, 50 μM 2-mercaptoethanol, 1% penicillin-streptomycin, 1% HEPES and 10% heat-inactivated FBS (HI-FBS) (Biowest, South American origin, VWR).

The analyzed human PBMC samples were collected from day 140 post 1st immunization. PBMCs were thawed using prewarmed AIM-V medium (Thermo Fisher Scientific, Gibco # 12055-091) supplemented with 1% penicillin-streptomycin, 20 μg/ml DNase and 10% HI-FBS. After centrifugation at 400 × *g* for 10 min. at room temperature cells were resuspended in media without DNase and rested overnight in the incubator at 37 °C, 5% CO₂. Subsequently cells were counted by an automated cell counter and number of dead cells subtracted.

### Cell culturing for cytokine detection in supernatants
Single mouse splenocyte cell suspensions were adjusted to $2 \times 10^5$ cells/well and stimulated in triplicates with CTH522, CTH523 and CTH518 at a final concentration of 5 μg/ml and CTH522 overlapping peptides at a final concentration of 10 μg/ml. After 72 h of incubation at 37 °C, 5% CO₂, the culture supernatants were harvested and stored at −20 °C. The amounts of secreted IFN-γ and IL-17A were determined by ELISA.

Human PBMC concentrations were adjusted to $1.25 \times 10^5$ cells/well in DNase free AIM-V medium supplemented with 1% penicillin-streptomycin and 10% HI-FBS and stimulated in triplicates with CTH522 (15 μg/ml), CTH523 (5 μg/ml), CTH518 (5 μg/ml) and CTH522 overlapping peptides at a final concentration of 10 μg/ml. After 5 days of incubation at 37 °C, 5% CO₂, the culture supernatants were harvested and stored at −20 C. The amounts of secreted IFN-γ and IL-17A were determined by ELISA. Supernatants were also analysed with the MSD V-plex kit; Proinflammatory panel 1 (See Supplementary Fig. 1 for more details).

### Flow Cytometry
Mouse cells were stimulated for 1 h in the presence of CTH522 (5 μg/ml) or without antigen in media containing costimulatory anti-CD28-purified (1 μg/ml) (BD Bioscience, clone: 37.51 cat. #553295) and anti-CD49d-purified (1 μg/ml) (BD Bioscience, clone: 9C10-MFR4.B, cat. #553313). Brefeldin A (SigmaAldrich; B7651-5mg) was added to each well to a final concentration of 10 μg/ml. After 6 h of incubation at 37 °C, the cells were kept at 4 °C until staining.

Surface staining: Cell suspensions were Fc-blocked with anti-CD16/CD32 antibody (BD Bioscience, clone 2.4G2, cat. #553142, 1:100) for 10 min. at 4 °C and stained for surface markers diluted in 50% brilliant stain buffer (BD Horizon, cat. #566349) as indicated and fixable viability dye Viability-eFlour780 (eBioscience, #65-0865-14, 1:500) at 4 °C for 20 min. The cells were then fixed and permeabilized using Cytofix/Cytoperm Solution kit (BD Bioscience, #554714) as per manufactorer's instructions and intracellular cytokine staining (ICS) for IFN-γ, IL-17A, TNF-α, IL-2 was performed at 4 °C for 30 min. Cells were stained with combinations of the following anti-mouse antibodies conjugated to fluorochromes (company, clone, catalog, dilution): α-CD4-BV510 (BioLegend, cRM4.5, #100559, 1:400), α-CD44-Alexa fluor 700 (Biolegend, IM7, #103026, 1:150), α-CD8-BV421 (Biolegend, 53-6.7, # 100738, 1:200), α-CD3-BV605 (BD Bioscience, 145-2C11, #563004, 1:100), α-CD19-BV786 (BD Bioscience, 1D3, #563333 1:300), α-IL-2-APC (eBioscience, JES6.5H4, #17-7021-82, 1:200), α-IFNγ PE-Cy7 (eBioscience, XMG1.2, #25-7311-82, 1:200), α-TNF-PE (eBioscience, MP6-XT22, #12-7321-82, 1:200), α-IL-17A-PerCP-Cy5.5 (eBioscience, eBio17B7, #45-7177-82, 1:200). The stained cells were analyzed using a Flow cytometer (BD LSRFortessa, BD Bioscience) and FlowJo Software (version 10). Non-specific background cytokine positive events from paired non-CTH522 stimulated cells were subtracted from each

Boolean gate individually. See Supplementary Fig. 11 for gating strategies in Spleen and GT.

### IFN-γ and IL-17A sandwich ELISA
**Mouse IFN-γ and IL-17A ELISA.** Maxisorp plates (Nunc, Roskilde Denmark) were coated with 100 μl rat anti-mouse IFN-γ (BD Pharmingen cat. #551216, clone R4-6A2) or rat anti-mouse IL-17A (Biolegend, cat. #506902, clone TC11-18H10.1) at a concentration of 1 μg/ml in carbonate buffer (SSI Diagnostics 24203), incubated overnight at 4 °C followed by blocking for 2 h in 1xPBS + 2% skim milk powder (SM) (Natur Drogeriet). After a 1x washing step with 1xPBS + 0.2%Tween-20 (washing buffer), harvested supernatants and either a recombinant IFN-γ standard (BD Pharmingen, cat. #554587) or a recombinant IL-17A standard (Biolegend, cat. #564101) were diluted in 1xPBS + 2%BSA, transferred to ELISA plates and incubated overnight at 4 °C or 2 h at room temperature. Plates were washed 3 times and 100 μl biotin conjugated rat-anti-mouse IFN-γ (BD Pharmingen, cat. #554410, clone XMG1.2, 0.1 μg/ml in 1xPBS + 1%BSA) or 100 μl biotin conjugated anti-mouse IL-17A (BioLegend, cat. #507002, TC11-8H4, 0.25 μg/ml in 1xPBS + 1%BSA) were added to each well and incubated for 1 h at room temperature. Plates were washed 3 times in washing buffer and 100 μl HRP-streptavidin (BD Pharmingen, cat. #554066, 1:5000) solution was added to each well, incubated for 30 min. at room temperature. After 5 washing steps, 100 μl TMB-PLUS (Kem-En-TEC, Taastrup, Denmark) was added to each well and the reaction was stopped with 100 μl 0.2 M H₂SO₄ after 30 min. OD (450-620 nm) was read using an ELISA reader. Mean responses in triplicate control wells were subtracted the mean value of antigen stimulated wells and values below 0 were assigned the value 0.5.

**Human IFN-γ ELISA.** Maxisorp plates (Nunc, Roskilde Denmark) were coated with 50 μl anti-Human IFN-γ (Thermo Fisher Sci, Invitrogen, cat. #M700A, clone 2 G, 2 μg/ml) and incubated overnight at 4 °C. Next day plates were washed once in 1xPBS + 0.05% Tween-20 (washing buffer) and blocked for 2 h in 1% BSA + 5% Trehalose (Sigma cat. #A8022) in 1xPBS. 100 μl diluted (dilution buffer, 1%BSA + 0.05% Tween-20 in 1xPBS) supernatants and an IFN-γ standard (Thermo Fisher Sci., Invitrogen, cat. #RIFNG50) were transferred to the ELISA plates and incubated overnight at 4 °C. Plates were washed 4 times and 100 μl biotin conjugated anti-human IFN-γ (Thermo Fisher Sci., Invitrogen, cat. #M701B, clone B133.5, 25 ng/ml in dilution buffer) were added to each well and incubated for 2 h at room temperature in the dark. Plates were washed 4 times and 100 μl HRP-streptavidin (BD Pharmingen, cat. #554066, 1:20.000) solution was added to each well and incubated for 30 min. at room temperature. After 6 washing steps, 100 μl TMB-PLUS (Kem-En-TEC, Taastrup, Denmark) was added to each well and the reaction was stopped with 100 μl 0.2 M H₂SO₄ after 30 min. OD (450-620 nm) was read using an ELISA reader. Median responses in control wells were subtracted the median value of antigen stimulated wells and values below 0 were assigned the value 0.5.

**Human IL-17A ELISA.** Human IL-17A was detected using a Human IL-17A ELISA kit (Thermo Fisher Sci., Invitrogen, cat. #BMS2017) according to the manufacturer's protocol except for the detection step were 100 μl TMB-PLUS (Kem-En-TEC, Taastrup, Denmark) was added to each well and the reaction was stopped after 10 min. with 100 μl of 0.2 M H₂SO₄. OD (450-620 nm) was read using an ELISA reader. Median responses in control wells were subtracted the median value of antigen stimulated wells and values below 0 were assigned the value 0.5.

### ELISA for antigen-specific antibodies in serum and vaginal washes
**Mouse: IgG, IgG1, IgG2a/b/c ELISA.** Blood was collected post vaccination and serum was isolated after 10 min. of centrifugation at 10,000 g. Vaginal wash samples were collected by flushing the vagina

with 100 μl of sterile 1 x PBS and samples stored at −80 °C until analysis. Before dilution, the vaginal wash samples were treated with 25 μg/ml Bromelain (Sigma-Aldrich). Maxisorp plates (Nunc, Roskilde Denmark) were coated with either recombinant antigens (1 μg/ml), peptides (10 μg/ml) or *C.t.* serovars (10 μg/ml) overnight at 4 °C, followed by blocking for 2 h in 1 x PBS + 2 % BSA for IgG. The serum and vaginal wash samples were serially diluted in 1 x PBS + 1% BSA for IgG or added to coated plates in a 1:200 dilution (B-cell epitope mapping). After washing, HRP-conjugated rabbit anti-mouse IgG (Thermo Fisher Sci., Invitrogen, cat. #61-6520, 1:2000), HRP-conjugated goat anti-mouse IgG1 (Southern Biotech, cat. #1070-05, 1:16000), HRP-conjugated rabbit anti-mouse IgG2a (Life Technologies, cat. #61-0220, 1:5000), HRP-conjugated goat anti-mouse IgG2b (Thermo Fisher Sci., Invitrogen, cat. #M32407,1:4000) and HRP-conjugated goat anti-mouse IgG2c (Thermo Fisher Sci., Invitrogen, cat. #PA1-29288, 1:20000) were added. Antigen specific antibodies were detected using TMB-PLUS (Kem-En-TEC, Taastrup, Denmark). The reaction was stopped with $H_2SO_4$ and OD (450–620 nm) was read using an ELISA reader. Results were presented either as titration curves, as absorbance (Abs) at one dilution after background subtraction (values below 0 were assigned the value 0, or as Arbitrary Elisa Units (AEU) calculated from an internal standard (serum pool from CTH522 vaccinated mice) using a five-parameter logistic curve using the package 'drc' in R.

**Human: Peptide and C.t surface specific IgG ELISA.** 96-well Maxisorp plates (Nunc, Roskilde, Denmark) were coated with 50 μl of 20–21-mer peptides (10 μg/ml) or *C.t.* serovars (10 μg/ml) overnight at 4 °C followed by blocking for 2 h in 1 x PBS with 2% SM. Serum samples were either serially diluted in 1 x PBS with 1% SM or added to the plate at a fixed 1:200 dilution (B cell epitope mapping). After washing (1xPBS + 1%SM), HRP-conjugated rabbit anti-human IgG (Agilent, Dako, cat. #P021402-02, 1:8000) were added. Antigen specific antibodies were detected using TMB-PLUS (Kem-En-TEC, Taastrup, Denmark). The reaction was stopped with $H_2SO_4$ and OD (450-620 nm) was read using an ELISA reader. Results were presented as either absorbance (Abs) at one dilution (1:200, B cell epitope mapping) or as AEU calculated from an internal standard established by combining samples with a positive response to *C.t.* SvD. The titers were calculated based on a five-parameter logistic curve using the package 'drc' in R.

**Human: CTH522 specific IgG, IgG1 and IgG3 ELISA.** 96-well Maxisorp plates (Nunc, Roskilde, Denmark) were coated with 50 μl/well of CTH522 diluted to 0.125 μg/ml in carbonate buffer pH 9.6. The plates were incubated for 2 h at room temperature or overnight at 2–8 °C. After incubation, the plates were washed 8 times with wash buffer (PBS pH 7.2 with 1% Tween 20) using an automated plate washer. The reference serum pool and samples were diluted in dilution buffer (Wash buffer + 1% BSA). The sera from the participants were tested by two-fold serial dilution in parallel with the reference serum pool and the plates were incubated for 2 h at room temperature or overnight at 2–8 °C. The plates were washed and HRP-labelled a) Rabbit anti-human IgG (Agilent, Dako, #P021402-02, 1:3000), b) Mouse anti-human IgG1 (Thermo Fisher Sci., Invitrogen, clone HP6069, cat. #A-10648, 1:200) or c) Mouse anti-human IgG3 (Thermo Fisher Sci., Invitrogen, clone HP6047, cat. #05-3620, 1:200) diluted in dilution buffer was added to wells and plates were incubated for 1 h at room temperature. The plates were washed and substrate (OPD dissolved in a citrus buffer pH 5.5) was added. The plates were incubated in the dark for 30 min before the reaction was stopped with 100 μl 1 M $H_2SO_4$. OD (492 nm) was read using an ELISA reader. A reference line approach on log-log transformed data was used to calculate the concentration of anti-CTH522 IgG antibodies in the serum samples using the reference serum pool as the calibrator. The samples were repeated if the dilution series had less than 3 points within the OD spectrum confined by the reference.

**Human: CTH522 specific IgG2 and IgG4 ELISA.** Due to low content of both anti-CTH522 IgG2 and CTH522 IgG4 in the internal reference serum pool, the reference line approach was changed to a fixed dilution assay testing samples from same donor on the same plate. Plates were coated with 50 μl/well of CTH522 diluted to 0.125 μg/ml in carbonate buffer pH 9.6. The plates were incubated for 2 h. at room temperature or overnight at 2–8 °C. After incubation, the plates were washed 8 times with Wash buffer (PBS pH 7.2 with 1% Tween 20). All sera from the participants were tested as triplicates in a 1:10 dilution. Plates were incubated for 2 h. at room temperature or overnight at 2–8 °C. The plates were washed and HRP-labelled (a) Mouse anti-human IgG2 (Thermo Fisher Sci., Invitrogen, clone HP6014, cat. #050520, 1:25) or (b) Mouse anti-human IgG4 (Thermo Fisher Sci., Invitrogen, clone HP6025, cat. #A10564, 1:50) was added to wells. Hereafter the plates were incubated for 1 h at room temperature and the substrate reaction was run as describe above. The mean of triplicates values was multiplied with the dilution factor to obtain an AEU.

### In vitro and in vivo neutralization

**In vitro neutralization assay.** The assay was performed in Hamster kidney cells (HaK) (ATCC CCL-15™) and was done essentially as published by ref. [46]. Briefly, HaK cells were maintained in RPMI 1640 supplemented with 1% (vol/vol) L-glutamine, 1% non-essential amino acids, 1% sodium pyruvate, 70 μM 2-mercaptoethanol, 10 μg/ml gentamicin, 1% HEPES and 5% heat inactivated fetal bovine serum at 37 °C, 5% $CO_2$. Cells were grown to confluence in 96-well flat-bottom microtiter plates (Nunc). The different *C.t.* stocks were diluted to a predetermined concentration in SPG buffer and mixed with heat-inactivated (56 °C for 30 min) and serially diluted serum. The mixture was incubated for 45 min. at 37 °C and inoculated onto HaK cells in duplicates or triplicates. After 2 h of incubation at 36 °C on a rocking table the mixture was removed and the cells were and further incubated 24 h at 37 °C, 5 % $CO_2$ in culture media containing 0.5% glucose and Cycloheximide (1:1000). The cells were fixed with 96% ethanol and inclusions were visualized by staining with polyclonal rabbit anti-rCT043 or rabbit anti-rCT110 serum (produced in our lab), followed by Alexa 488-conjugated goat anti-rabbit immunoglobulin (1:500-1:1000) (Thermo Fisher Sci., Invitrogen, cat #A11008). Cell staining was done with Propidium Iodide (Thermo Fisher Sci., Invitrogen). IFU were enumerated by fluorescence microscopy using an automated cell imaging system (ImageXpress Pico automated Cell imaging system (Molecular Devices, San Jose, California, USA and CellreporterXpress software) counting 25–90% of each well or by manual counting. The neutralization was calculated as percentage reduction in mean IFU relative to control serum. A serum dilution giving a 50% or greater reduction in IFU relative to the control was defined as neutralizing. The serum dilution giving a 50% reduction in IFU was named reciprocal 50% neutralization titer ($NT_{50}$). $NT_{50}$ values were calculated based on a five-parameter logistic curve using the package 'drc' in R. For samples where no titer could be calculated, the samples were assigned a titer of half the lowest dilution tested; for experiments with human serum: 10, for experiments with mouse serum Fig. 2f: 16, Fig. 2h:12.5 or 25 and Fig. 7b: 8.

**In vitro neutralization assay using the endocervical cell line.** The cell line End1/E6E7 (ATCC CRL-2615™)[36] was maintained in Complete Keratinocyte-Serum Free Medium (K-SFM) (Keratinocyte-Serum Free Medium (Thermo Fisher Sci., cat. #17005-042) supplemented with 0.4 mM $CaCl_2$, 0.05 mg/ml Bovine Pituitary Extract (BPE), 0.1 ng/ml human recombinant Epidermal Growth Factor (rEGF), 10 μg/ml Gentamicin and 1% vol/vol HEPES. Cells were incubated at 37 °C, 5% $CO_2$ and the neutralization assay was performed as described above with the only exception that cells were incubated 24 h at 37 °C, 5% $CO_2$ in 100 μl Complete K-SFM containing 0.5% glucose and Cycloheximide

(1 μg/ml). Participants with $NT_{50} > 200$ in HaK cells were tested using End1/E6E7 cells.

**Competitive inhibition of neutralization.** In the competitive inhibition of neutralization assay purified serum from CTH522 vaccinated mice and control mice were diluted 1:200 in SPG buffer and pre-incubated for 45 min. at 37 °C with 0.5 mg/ml CTH522, CTH523, CTH518 and $VD4_{6-22}^{D/E/F/G}$ or SPG buffer alone prior to 45 min. of incubation at 37 °C with equal volume of *C.t.* SvD, E, F, and G. The mixtures were inoculated onto a HaK cell monolayer in duplicate and the assay was performed as described above.

**In vivo neutralization.** Human (a serum pool of 7 CTH522 vaccinated participants with *C.t.* SvD $NT_{50} > 200$, and a serum pool of 5 placebo participants) and mouse (a serum pool of CTH522 vaccinated mice and a serum pool of control mice) serum pools were heat-inactivated, sterile-filtered and diluted 1:2 and 1:4 with a fixed concentration of *C.t.* SvD. After 30 min. at 37 °C, depo-provera treated C3H/HeN mice were infected with 10 μl of the inoculum (a total of $1.5 \times 10^3$ IFU/mouse), swabbed at day 3 and 7 post infection, and IFU were determined as described below.

## Vaginal challenge and bacterial burden

Ten and three days before *C.t.* SvD challenge, the oestrous cycle was synchronized by injection of 2.5 mg Medroxyprogesteronacetat (Depo-Provera, Pfizer, Ballerup, Denmark), increasing mouse susceptibility to chlamydial infection by prolonging dioestrus. *C.t.* SvD in concentrations of $1.5 \times 10^3$–$1 \times 10^5$ IFU/mice *C.t.* SvD in 10 μl SPG buffer was inoculated i.vag. At several days post infection (PID) mice were swabbed vaginally. To determine protection against ascending infection in immunocompetent B6C3F1 mice, the GTs were removed at PID21-22 and the uterine horns (middle GT; mGT) and upper genital tract tissue (uGT) comprising the fallopian tubes and ovaries were homogenized (in one experiment the horns were swabbed, Fig. 6g, Exp. 3) using a gentleMACS™ Dissociator (Miltenyi Biotec). The homogenates were placed in Eppendorf tubes with glass-beads and stored at −80 °C until analysis. Before use they were vortexed for 1 min. Swab samples were collected in 0.6 ml SPG buffer with glass-beads, vortexed for 1 min. and stored at −80 °C until analysis. The infectious load was assessed in GT tissue or swab material by infecting 48 plate wells seeded with McCoy cells with the GT tissue or swab material. The McCoy cells were maintained in RPMI 1640 supplemented with 1% L-glutamine, 1% non-essential amino acids, 1% sodium pyruvate, 70 μM 2-mercaptoethanol, 10 μg/ml gentamicin, 1% HEPES and 5% heat inactivated fetal bovine serum at 37 °C, 5% CO₂. Cells were grown to confluence in 48-well flat-bottom microtiter plates (Costar, Corning, NY, USA). The cells were infected by 1 h of centrifugation at 750 g. After 2 h of incubation at 37 °C the wells were aspired and incubated 24 h at 37 °C, 5 % CO₂ in 100 μl culture media containing 0.5% glucose and Cycloheximide (1 μg/ml). After 24 h of incubation wells were aspired. Inclusions were visualized by staining with an in-house polyclonal rabbit anti-MOMP serum, followed by an Alexa 488-conjugated goat anti-rabbit immunoglobulin (Thermo Fisher Sci., Invitrogen, cat. #A11008, 1:500–1:1000). Background staining was done with Propidium iodide (Invitrogen). IFU were enumerated by fluorescence microscopy either manually or by using an automated cell imaging system (ImageXpress Pico automated Cell imaging system) (Molecular Devices) counting 50% of each well. If no IFU were detected in the counted area, 100% of each well was counted manually. For the swab samples culture-negative mice were assigned a limit of detection of 4 IFU/mouse representing one IFU in the tested swab material (1/4 of the total swab material). To investigate the ability of the vaccine to protect against upper genital infection, all tissue homogenates as well as all swab materials were analysed and the results presented as culture positive samples out of total.

## Depletion of CD4⁺ and CD8⁺ T cells

C3H/HeN mice were depleted of CD4⁺ and CD8⁺ T cells by i.v. administration of monoclonal rat anti-mouse CD4 (Bio X Cell clone GK1.5, cat. #BE0003-1) and rat anti-mouse CD8 (Bio X Cell clone YTS169, cat. # BE0117) diluted in Dilution buffer pH 7.0 (Bio X Cell, cat. #IP0070). At depletion injections days PID-11, −6, −3 and 1 mice received 200 μg CD8 and 400 μg CD4 antibodies and at depletion injections PID 8, 15, and 22 they received 100 μg CD8 and 200 μg CD4 antibodies. The depletion status in blood and GT tissues were verified by flow cytometry (Supplementary Fig. 6).

## Purification of antibodies and adoptive transfer of IgG to C3H/HeN mice

Blood was collected from CTH522 vaccinated and control B6C3F1 mice (Exp.1 $n = 100$, Exp 2. $n = 150$) and serum isolated by centrifugation at 10,000 g for 10 min. In Exp. 1 mice were vaccinated 3 times with 14 days interval with 25 μg CTH522/CAF®01 and serum were isolated and pooled 14 days post last vaccination. In Exp. 2 mice were vaccinated 3 times with 14 days interval with 10 μg CTH522/CAF®01 followed by a boost with 25 μg CTH522 without adjuvant. Sera were isolated and pooled 10 days post last vaccination. The pools of serum were filtered through a 0.45 μm Minisart® syringe filter (Sartorius), diluted 1:2 by PBS and IgG were purified on a 5 mL HiTrap™ Protein G HP column (GE-healthcare Bio-Sciences AB) according to the manufacturers protocol. Collected fractions were dialyzed overnight against PBS using a Slide-A-Lyzer™ Dialysis Cassette (Thermo Fisher Scientific), filtered through a 0.22 μm Minisart® syringe filter (Sartorius), IgG concentration quantified by Nano-Drop™ 2000, and the purified IgG stored at −20 °C until use. (Supplementary Fig. 8; SDS page of purified IgG from CTH522 vaccinated and control mice and CTH522 specific IgG1 and IgG2c responses of purified IgG). Purified IgG were transferred to recipient C3H/HeN mice by the intraperitoneal (i.p.) route. In Exp. 1 each recipient mice received either a total of 1 mg purified IgG from CTH522 vaccinated mice or 0.43 mg IgG from control mice at PID-3. In Exp.2 each recipient mice received either a total of 4.6 mg purified IgG from CTH522 vaccinated mice or 1.43 mg IgG from control mice at PID-3 and at PID15 they received an additional 1.44 mg IgG from CTH522 vaccinated or 0.06 mg IgG from control mice.

## Immunohistochemical staining

Genital tracts from mice were removed following euthanasia and fixed at room temperature in 4% formaldehyde (VWR chemicals) and paraffin embedded. Processing, sectioning and staining were done by the technical staff at BioSiteHisto (Finland). Four μm thick sections were collected on TOMO adhesive coated slides for immunohistochemical (IHC) staining. Detection of *C.t.* was performed with an in-house rabbit anti-MOMP polyclonal antibody and an HRP based detection system and a brown DAB chromogen. The IHC slides were counterstained with Mayer Hematoxylin (Merck). Slides were digitalized as WSI in Mirax format with 3DHistech Panoramic MIDI scanner (3DHistech Ltd.). Slides were visualized and analyzed using CaseViewer version 2.3. (3Dhistech Ltd.).

## Statistical analysis

GraphPad Prism v 9.3.1. and 10.0.2 was used for data handling, analysis and graphical visualizations. The statistical tests used are described in the relevant figure legends and p-values are shown either in the figure or in figure legends. A p-value above 0.05 was considered not significantly different. *$p < 0.05$, **$p < 0.01$, ***$p < 0.001$, ****$p < 0.0001$. ns, non-significant.

## Reporting summary

Further information on research design is available in the Nature Portfolio Reporting Summary linked to this article.

## Data availability
Data that support the findings of this study are presented in the manuscript and Supplementary Information. All data are provided in the Source data file provided with this paper. Source data are provided with this paper.

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

## Acknowledgements

The clinical phase I trial was supported by the European Commission through the ADITEC consortium contract (FP7-HEALTH-2011.1.4-4-280873, R.S.) and The Innovation Fund Denmark (069-2011-1). We appreciate the excellent technical assistance provided by Lene Rasmussen, Anne Bjørlig, Sara Knudsen, Vivi Andersen, Mitra Pourjam, and

the staff at the experimental animal facility at Statens Serum Institut. We thank Emma Kathrine Lorenzen for scoring the pathology of the genital tract tissue. We thank Martin J. Holland, Department of Clinical Research, London School of Hygiene & Tropical Medicine for providing SvA/2497 and SvB/Tunis-864.

## Author contributions

A.W.O. designed and conducted experiments, performed data analysis, generated figures and tables and wrote the manuscript. I.R. contributed with materials and reagents, conducted experiments, and revised the manuscript. C.S.J. conducted experiments, performed data analysis, generated figures, and revised the manuscript. H.M.C. conducted the phase 1 clinical trial and revised the manuscript. M.P.K. conducted experiments. J.D. contributed to the interpretation of the results and revised the manuscript. R.S. conducted the phase 1 clinical trial and revised the manuscript. F.F. contributed to the design of the experiments, the interpretation of the results, the writing of the manuscript and supervised the project. All authors read and approved the final manuscript.

## Competing interests

A.W.O., I.R. and F.F. are co-inventors of a patent (WO2014146663A1) relating to *C.t.* vaccines. All rights have been assigned to Statens Serum Institut, a Danish not-for-profit governmental institute. The remaining authors declare no competing interests.
