## [Peer Review File · Nature Communications]

REVIEWER COMMENTS

Reviewer #1 (Remarks to the Author):

In this manuscript Olsen et al. present a comparative analysis of the immune responses to a Chlamydia vaccine (CTH522/CAF01) in humans and mice. Following vaccination, similar antibody functionality and Th1/Th17 cytokine responses were found both in humans and mice. PBMCs from humans and splenocytes from mice vaccinated with CTH522/CAF01, were found to secrete high levels of IFN- γ and IL-17 when stimulated with CTH522. The authors also identified T-cell epitopes in MOMP that were previously described in Chlamydia infected patients and are well-conserved among the various C. trachomatis serovars. These common immune responses correlated with long-term protection against vaginal shedding and upper genital tract pathology in the mouse model.

The findings reported in this manuscript are important since provide support for the efficacy of CTH522/CAF01 vaccine, define immunological responses that correlate with vaccine induced protection and also because they validate the mouse model for testing Chlamydia vaccines.

Issues to address

A recent publication showed that the CTH522/CAF01 vaccine does not induce significant protection in non-human primates against a C. trachomatis genital challenge. The results by Olsen et al. in the mouse model appear to contradict those results. The data of the non-human primates studies needs to be presented and discussed in this paper.

CTH522 was used to stimulate T-cells from vaccinated humans and mice to produce cytokines. Was data collected using Chlamydia EB to stimulate T-cells rather than CTH522? That data will be more relevant than the one presented here.

Although levels of IFN- γ in T-cell supernatants are known to correlate with protection the role of IL-17A is a little bit more controversial. Th17 and IL-17A have anti-infective activity but they also contributes to Chlamydia induced inflammatory pathology. This needs to be discussed. Also, data for other cytokines/chemokines from vaccinated humans and mice could be presented.

The authors make the important point that humans will probably be vaccinated before their sexual debut. However they used 6–8-week-old mice to perform these studies. Considering that by that age

most of the mice are sexually and immunologically more mature than 3–6-week-old mice it may be important to test the vaccine in younger animals to better mimic the age at which most humans will be vaccinated.

Throughout the manuscript equivocal sentences such as, “The experiment was repeated with similar results” and “The analysis was repeated with similar results”, cast doubts of whether the results were really similar and therefore, should be analyzed with the previous experiment, or they are suggesting that the results were not really reproducible. In page 11 the results of two in vivo neutralization experiments are reported. For experiment 1 the P value was 0.0143 and for the second experiment it was 0.0993. Fig. 6f shows data from three similar experiments and the results are presented individually and as an aggregate. The different forms of analyzing and presenting the data are confusing and concerning.

The histological images in Fig. 5 are uninterpretable due to their size and magnification.

In the long-term protection experiment antibody titers were measured in serum showing that overtime, there was a considerable decline in titers (67-fold). However the results of the in vitro neutralization are not presented. This information is critical since they will help to clarify whether or not neutralizing antibodies play a significant role in protection against a genital challenge. Similarly, neutralizing antibody titers are not reported after the challenge.

In Fig. 4c it seems like the neutralization of the IFU was short-lived. Will the authors clarify these results? They state that, “The experiment was repeated with similar results”. If that is the case analyzing together the results of the experiments should improve the P values.

Can the authors explain the biological and/or functional significance of having hydrosalpinxes of different sizes? Can they explain the significance of these differences in size? What scale was used in the Y axis of Suppl. Fig. 4?

Based on the results shown in Fig. 3 it will appear than only VD4 induces neutralizing antibodies. Where the other three VDs also tested?

Are B6C3F1 mice known to produce both IgG2a and IgG2c antibodies?

The neutralizing ability of serum from vaccinated mice was tested in animals depleted of CD4+ and CD8+ T-cells. This is an interesting model however it does not evaluate what neutralizing antibodies will do in

vaccinated individuals with normal cell mediated immunity as will happen for the majority of vaccinated humans.

The manuscript needs to be carefully searched for formatting errors. For example, in Fig. 4a, the legends for the Y axis are reversed. In Fig. 6 the g) of the legend should be f).

Reviewer #2 (Remarks to the Author):

Chlamydia trachomatis infection is the most common sexually transmitted disease; however, an efficacious vaccine is currently lacking. The manuscript by Dr. Olsen is reporting immunological characterization of vaccine candidate CTH522/CAF®01 in mice model and human serum. Its safety and immunogenicity from first phase 1 study was previously reported (PMID: 31416691). In this manuscript, the authors reported CTH522/CAF®01 vaccine candidate can elicit effective antibody and T cell response and provide partial protection in infectious loads and pathology which could lead to further clinical trial investigate, however, several critical issues need to be addressed to make it convincing.

1. CTH522/CAF®01 was designed as a multivalent vaccine against serovar D, E, F and G, therefore, neutralization assays against other serovars (including E, F and G) in vitro need to be performed in Figure 3a, Figure 4b and Figure 4d.
2. In Figure 4b, serum from CTH522 exhibited strong in vitro neutralization capacity, however, the in vivo neutralization assay at day 7 post infection did not reflect in vitro results. The authors need to address it, meanwhile, what are the in vivo neutralization assay results at day 14, 21 and 28 post infection.
3. In Figure 5c, do the mice culture negative at day 3 post infection remain negative though the whole culture period in Exp.1 and Exp.2?
4. In lines 221-223, authors stated "In mice with sustained high levels of IFU in the vaginal tract, throughout the investigated time period, the inflammatory response progressed into fluid filled hydrosalpinx at PID29 (Fig. 5e, panel 6). In Depl.+ CTH522 IgG group, what is the percentage of infected mice with high IFU developing pathology in Oviduct/ovary?"
5. In Figure 5e, authors need to characterize inflammatory cells in the tissue as T cells were depleted.
6. In Figure 6f, what is the percentage of infected mice from CTH522 group developing pathology in Oviduct/ovary?

7. Native MOMP provide protection against infectious burden but not pathology in a nonhuman primate trachoma model has been previously reported (PMID: 19494332), which should be referenced.

Reviewer #3 (Remarks to the Author):

Immune signature of the Chlamydia vaccine CTH522/CAF1[®]01 translates from mouse to human and induces durable protection. Olsen et al.

General: This is a well written paper that describes the T cell and antibody immune responses to a MOMP peptomer vaccine adjuvanted with CAF01 after delivery in human females and mice. Confirmation of the ability of the vaccine to significantly reduce bacterial burden in female mice challenged with human serovar D of *C. trachomatis* is confirmed (previously published by this group) and additional data indicate reduced ascension and protection from oviduct pathology in vaccinated mice. Data indicate similar T cell responses are induced in humans and mice after vaccination with CTH522/CAF01. Appropriate controls are included and sufficient numbers of mice and repeat experiments provide evidence of rigor and reproducibility of the results. The long-term protection studies are particularly impactful.

Statements related to the protective role of antibody should be deemphasized and qualified in the discussion. Many T cell depleted mice continue to develop upper genital tract infection and oviduct disease despite development of high levels of CTH522-specific antibody. Further, no studies were conducted that remove the effects of antibody and test for reduced protection in the absence of antibody, e.g., vaccination and challenge of B cell deficient mice, or adoptive transfer of immune T cells into RAG-deficient mice – such studies would complement the current data. Although these additional studies are not required – since they were not performed, and human *C. trachomatis* inoculated into mice produces an infection of reduced intensity and duration compared to *C. muridarum* inoculated into mice due to host species-specific chlamydia-specific effects of interferon-gamma – the role of antibody may be exaggerated in this model. Since the infection is already attenuated using this *C. trachomatis* in female mice model – the ability of antibody to inhibit infection further may be accentuated.

Title: The title suggests the vaccine has been demonstrated to induce durable protection in humans. The title should be rewritten to better indicate what has been demonstrated in mice and what has been demonstrated in humans.

Abstract: “Interestingly” – could be deleted – let the reader decide what is interesting.

Introduction:

1. Fibrotic blockage of the tubal and resulting infertility or ectopic pregnancy – should be fibrotic blockage of the fallopian tubes or fibrotic blockage of oviducts.

2. Typos –

Overall, the outcomes of our comparative analysis of CTH522 induced immune responses in mice and humans are promising in relation to the protective efficacy of CTH522 as a Chlamydia vaccine for humans that is likely to be introduced alongside the HPV vaccine and therefore needs to be sustained for at least a decade.

CTH522 is a recombinant engineered version of the Major Outer Membrane Protein (MOMP) from C.t. MOMP and holds four highly variable domains VD1 to VD4, spaced between serovar conserved segments (CSI-CSV).

To localize and compare the breadth of the T cell epitope regions of CTH522, mouse splenocytes and human PBMCs were further stimulated...

Results:

1. Preincubation of chlamydia in high levels of antibody-containing serum for an hour is likely very different than antibody interactions with CT EBs in cervicovaginal mucus or male urethral secretions/ejaculate. This should be mentioned in the discussion as a limitation of the experiments in immunologically normal mice that examined the role of antibody. Should also remark that CT infection is compromised in mice due to host-species-specific immune evasion and thus the model used could overestimate protective effects of antibody (see above).

2. The authors tend to overstate impact of antibody – In experiments that used T cell depleted mice - “Ten out of 30 mice (sum of both experiments) receiving CTH522 specific antibodies had a marked effect of the transferred antibodies, with no detectable bacteria at PID7.” The alternative– is that in 20 out of 30 mice minimal effect was observed – this should be reported. e.g., despite high levels of antibody – 20 of 30 mice continued to exhibit burden levels that were not significantly different than controls.

Discussion:

1. The authors should discuss that the immunization failed to neutralize serovar H, I Ia, and J despite inclusion of conserved regions of MOMP in the vaccine construct.
2. Line 374 – “vaginal tract” – strange terminology- - Chlamydia don’t infect the vagina – they infect the exocervix and endocervix and ascend up. The antibodies are protecting from cervical infection.
3. Otherwise – very nice balanced discussion of roles of CMI and antibody.

Point by point response to Reviewers comments

REVIEWER COMMENTS

Reviewer #1 (Remarks to the Author):

In this manuscript Olsen et al. present a comparative analysis of the immune responses to a Chlamydia vaccine (CTH522/CAF01) in humans and mice. Following vaccination, similar antibody functionality and Th1/Th17 cytokine responses were found both in humans and mice. PBMCs from humans and splenocytes from mice vaccinated with CTH522/CAF01, were found to secrete high levels of IFN- γ and IL-17 when stimulated with CTH522. The authors also identified T-cell epitopes in MOMP that were previously described in Chlamydia infected patients and are well-conserved among the various *C. trachomatis* serovars. These common immune responses correlated with long-term protection against vaginal shedding and upper genital tract pathology in the mouse model.

The findings reported in this manuscript are important since provide support for the efficacy of CTH522/CAF01 vaccine, define immunological responses that correlate with vaccine induced protection and also because they validate the mouse model for testing Chlamydia vaccines.

Issues to address

1. A recent publication showed that the CTH522/CAF01 vaccine does not induce significant protection in non-human primates against a *C. trachomatis* genital challenge. The results by *Olsen et al.* in the mouse model appear to contradict those results. The data of the non-human primate studies needs to be presented and discussed in this paper.

We have now inserted a paragraph discussing the non-human primate results. Page 20 l. 412.

“Similarly, to mice and humans, Lorenzen et al. (Lorenzen, 2022) found that CTH522/CAF[®]01 vaccinated Cynomolgus macaques established a strong and neutralizing antibody response in combination with a multifunctional CD4⁺ T cell response. The vaccine did not significantly accelerate clearance of infection compared to the control animals possibly due to the very high challenge dose of 5x10⁷ IFU of C.t. SvD/animal needed to establish infection in the NHP model, a dose that is far from the 675–15,920 copies/mL found in semen of infected patients (Al-Mously, 2009). We speculate, that this high dose model may call for an immune response encompassing all arms of the immune system, including CD8⁺ and CD4⁺ T cells, as well as neutralizing antibodies, in order to reduce the bacterial level”

2. CTH522 was used to stimulate T-cells from vaccinated humans and mice to produce cytokines. Was data collected using Chlamydia EB to stimulate T-cells rather than CTH522? That data will be more relevant than the one presented here.

EB stimulation was included in our mouse studies and results has now been inserted as supplementary Fig. 2. Here we see strong recognition of SvD EB's after CTH522 vaccination.

3. Although levels of IFN- γ in T-cell supernatants are known to correlate with protection the role of IL-17A is a little bit more controversial. Th17 and IL-17A have anti-infective activity but they also contribute to Chlamydia induced inflammatory pathology. This needs to be discussed. Also, data for other cytokines/chemokines from vaccinated humans and mice could be presented.

Discussion has been inserted on page 16 l. 325

“Th17 cells display a great degree of plasticity and although they can be pathogenic (Andrew, 2013), Th17 cells are important for the control of mucosal pathogens (Khader, 2009). Evidence support that Th17 cells have similar roles in humans and mice (Schmitt, 2015) and studies in mice show that Th17 cells have the ability to home to mucosal tissues, recruit neutrophils, antigen presenting cells and facilitate IgA secretion (Khader, 2009, Hirota, 2013). A recent study by Yount et al. (BioRxiv, 2023) highlighted that a T cell signature of central memory CD4 Th1, Th17, and Th17 cells are associated with reduced C.t. reinfection in a highly exposed cohort.”

We agree that more cytokines from vaccinated humans and mice could inform about the vaccine. We have inserted results for more cytokines in Supplementary Fig. 7 and found dominant levels of, IL-6, TNF- α and IL-13 besides IFN- γ and IL-17A. For the mouse, we refer to previous work from our group (Knudsen et al. 2016) on the adjuvant platform CAF[®]01 combined with recombinant MOMP. This study found corresponding results in PBMCs stimulated with recombinant MOMP with the cytokines IFN- γ , IL-6 and TNF- α as the dominating cytokines, when adjuvanted with CAF[®]01.

4. The authors make the important point that humans will probably be vaccinated before their sexual debut. However, they used 6–8-week-old mice to perform these studies. Considering that by that age most of the mice are sexually and immunologically more mature than 3–6-week-old mice it may be important to test the vaccine in younger animals to better mimic the age at which most humans will be vaccinated.

The CAF01 adjuvant system has previously been compared in adult and neonatal (1 week) immunization with a TB antigen and no differences in Th1 and Th17 responses were observed (doi: [10.1371/journal.pone.0005771](https://doi.org/10.1371/journal.pone.0005771), <https://doi.org/10.1038/s41467-022-31709-2>). Multifunctional Th1 cells were already elicited after a single vaccine dose and persisted at high levels for at least 6 months even after neonatal priming.

Regarding sexually maturity, mice are sexually mature between 5-8 weeks depending on strain, so all the mice will be sexually mature when challenged, even when vaccinating 3 weeks younger mice. It will therefore be difficult to fully mimic the situation in humans as the maturation rate in mice are so much faster, and we do not believe that vaccinating 3 weeks younger mice would have a huge impact on neither the immune response nor the infection.

5. Throughout the manuscript equivocal sentences such as, “The experiment was repeated with similar results” and “The analysis was repeated with similar results”, cast doubts of whether the results were really similar and therefore, should be analyzed with the previous experiment, or they are suggesting that the results were not really reproducible. In page 11 the results of two in vivo neutralization experiments are reported. For experiment 1 the P value was 0.0143 and for the second experiment it was 0.0993. Fig.

6f shows data from three similar experiments and the results are presented individually and as an aggregate. The different forms of analyzing and presenting the data are confusing and concerning.

We have now incorporated the replications in the figures, where it was meaningful or alternatively inserted experiments inducing similar results as supplementary figures (See below). It was not our intension to cast doubts. It is required from the journal that you state whether or not your experiments have been repeated. In all the crucial experiments, we decided to show them individually e.g Fig. 5. Exp.1 and Exp.2 as we found it more correct, since one of the experiments did not turn out to be significant (P=0.0993). In Fig 6g we got identical results in all three experiments and to enhance the number of samples for statistical analysis, we also presented a sum of the three experiments.

Fig. 1c and e are now a pool of three experiments (n= 12-13) instead of two (n=9-10), and we have deleted the sentence: "The experiment has been repeated with similar results. Fig. 6b is actually a fourth replication.

Fig. 1g. Mouse T cell epitope mapping: The replication has been incorporated into the figure, so we now have data from 12 mice instead of 6 and we have deleted the sentence: "The experiment has been repeated with similar results"

Fig. 3. The figure has been changed due to reviewers' comments. Results from neutralization of C.t. SvD with serum and IgG from the two individual experiments Exp.1 and Exp. 2 (Fig.5) have now been inserted individually. The blocking of neutralization figure has been changes and now shows blocking with different constructs. The neutralization assay has been repeated 3 times and results from all three determinations are presented.

Fig. 4c. The experiment was repeated, where the human and mouse serum was diluted 4 times in the SvD inoculum instead of 2. It is therefore not correct to pool them, and we have now inserted both experiments in Fig. 4c. They show similar results.

Fig. 4d. In vitro neutralization. The figure has been changed and neutralization of all 4 serovars are presented. Individual data points represent calculated mean neutralization of duplicate or triplicate readings.

Fig. 7f. Long-term protection with CTH522 has been done twice. In the other experiment two s.c. immunizations with CTH522/CAF01 were boosted with an intranasal vaccination without adjuvant. Doses of both 5 µg and 50 µg CTH522/CAF01 for the s.c. immunizations were tested. The kinetic of clearance 1-year post 3rd vaccination have been inserted as Supplementary figure 10 and the results are very similar to the one presented in the manuscript.

The histological images in Fig. 5 are uninterpretable due to their size and magnification.

The histological images were included to visualize the different degrees of pathology in relation to IFU's and to show that we could visualize chlamydia inclusions in the upper genital tract at PID29. We have enlarged the images slightly (now a 5x magnification). But we agree that it is not possible with the present magnification to see that neutrophils are part of the pathological response as we wrote.

We have therefore included a FACS analysis of the uGT at PID29 in the two groups (Supplementary Fig. 9). This shows that neutrophils are a large part of the uGT cells at PID29 and we have rewritten the sentence p 12.l 229 to:

"In contrast, mice that did not control the initial infection and had sustained high levels of IFU in the swabs throughout the investigated time period developed substantial oviduct inflammation with dense filling of debris in the lumen and visible EB inclusions (MOMP-positive staining) in both lumen and epithelial cells of

the oviducts (Fig. 5e panel 3-5). A FACS analysis of uGT tissue at PID 29 demonstrated the presence of a high percentages of neutrophils especially in mice receiving control antibodies, but also NK cells, macrophages, and dendritic cells (DC's) were present (Supplementary Fig. 9). Even severe pathology with fluid filled hydrosalpinx could be detected at PID29 (Fig. 5e panel 6).

6. In the long-term protection experiment antibody titers were measured in serum showing that overtime, there was a considerable decline in titers (67-fold). However, the results of the *in vitro* neutralization are not presented. This information is critical since they will help to clarify whether or not neutralizing antibodies play a significant role in protection against a genital challenge. Similarly, neutralizing antibody titers are not reported after the challenge.

This is a valid point. We have now run and inserted the neutralization titers over time (Fig. 7b).

We have inserted the results in the manuscript p14 | 278:

“In parallel with the decrease in serum antibody levels, the neutralizing capacity fell to a median NT₅₀ of around 100 in week 31 and to the detection limit of the neutralization assay at 1 year (Fig. 7b)”

and discussed the finding p21 | 432:

“Serum antibody levels decreased 11-fold after 6 month and 67-fold during the one-year period. This antibody trajectory likely means that over time the role of neutralizing antibodies in the immediate control of infection will decline, and instead rely on the memory T and B cell response (Fig. 7c). In fact, we measured a significant reduction in IFU levels already at PID 3, which became more pronounced at PID7 where the level of bacteria in the CTH522/CAF01 group was reduced from a median log₁₀ IFU of 4.1 (PID3) to 0.6 (PID7). We saw a recall antibody and CMI response following infection, slightly offset bacterial clearance, which could impact clearance of the infection and bacterial shedding”

7. In Fig. 4c it seems like the neutralization of the IFU was short-lived. Will the authors clarify these results? They state that, “The experiment was repeated with similar results”. If that is the case analyzing together the results of the experiments should improve the P values.

*The *in vivo* neutralization assay (Fig. 4c) was done to support our *in vitro* results in the HaK and the endocervix cell lines. In agreement with the *in vitro* neutralization assays we saw strong reduction in IFU's when EB's were coated with murine and human CTH522 specific antibodies compared to EB's coated with naive antibodies at early timepoints (PID3 and PID7) post challenge. The later outcome of the infection will depend primarily on whether the infection was completely cleared at PID3. Once intracellular chlamydia will grow since no CTH522 specific T cells and no additional CTH522 specific antibodies from the blood will reach the genital tract. Therefore, it is our interpretation that the *in vivo* neutralization assay reflects and supplements the *in vitro* neutralization assays.*

The experiment was repeated where the human and mouse serum was diluted 4 times in the SvD inoculum. These results are now inserted in Fig. 4c and show similar results

8. Can the authors explain the biological and/or functional significance of having hydrosalpinxes of different sizes? Can they explain the significance of these differences in size? What scale was used in the Y axis of Suppl. Fig. 4?

Having hydrosalpinxes of different sizes may have different functional significance depending on the cause and extent of the blockage. For one the size of the hydrosalpinx may reflect the severity of the damage to the fallopian tube and the surrounding tissues. A larger hydrosalpinx may indicate a more chronic or extensive infection or inflammation, or a more complete occlusion of the tube. A smaller hydrosalpinx may indicate a more acute or localized problem, or a partial occlusion of the tube.

Supplementary Fig.4 – now supplementary Fig. 7. Oviduct inflammation was scored from 0-4, ranging from normal to extreme inflammation and Oviduct dilation was scored from 0-5 ranging from no dilation to extreme dilation. It has now been inserted in figure legends. Each dot represents the sum of right and left oviduct from one mouse.

The figure should only be used to support that C.t. induced inflammation could be detected at PID29 – the time point used in Fig. 5

9. Based on the results shown in Fig. 3 it will appear than only VD4 induces neutralizing antibodies. Where the other three VDs also tested?

We have changed Fig. 3 to answer the question raised by the reviewer. Blocking of neutralization experiments has been improved to also include blocking of neutralization with CTH523 (VD1, VD2 and VD3 are all part of CTH523). Incubating the serum with CTH523 show no effect on the neutralizing capacity compared to incubating the serum with CTH522, CTH518 and a VD4 construct. We have inserted all the new results as Fig. 3c.

10. Are B6C3F1 mice known to produce both IgG2a and IgG2c antibodies?

Yes, the B6C3F1 mouse is a cross between C57Bl/6 (IgG2c) and C3H/HeN (IgG2a) and therefore have the genes for both subclasses.

11. The neutralizing ability of serum from vaccinated mice was tested in animals depleted of CD4+ and CD8+ T-cells. This is an interesting model however it does not evaluate what neutralizing antibodies will do in vaccinated individuals with normal cell mediated immunity as will happen for the majority of vaccinated humans.

We fully agree. This was done to describe the T cell independent protective role of antibodies. Protection in fully immunocompetent mice is presented in Fig. 6.

12. The manuscript needs to be carefully searched for formatting errors. For example, in Fig. 4a, the legends for the Y axis are reversed. In Fig. 6 the g) of the legend should be f).

We have changed Fig. 4a to simplify it. The error in Fig. 6 has been corrected. The legends were correct but the labeling on the figure was wrong, this has now been corrected in the figure and the text. In addition, we have corrected the formatting error pointed out by reviewer 3

Reviewer #2 (Remarks to the Author):

Chlamydia trachomatis infection is the most common sexually transmitted disease; however, an efficacious vaccine is currently lacking. The manuscript by Dr. Olsen is reporting immunological characterization of vaccine candidate CTH522/CAF[®]01 in mice model and human serum. Its safety and immunogenicity from first phase 1 study was previously reported (PMID: 31416691). In this manuscript, the authors reported CTH522/CAF[®]01 vaccine candidate can elicit effective antibody and T cell response and provide partial protection in infectious loads and pathology which could lead to further clinical trial investigate, however, several critical issues need to be addressed to make it convincing.

1. CTH522/CAF[®]01 was designed as a multivalent vaccine against serovar D, E, F and G, therefore, neutralization assays against other serovars (including E, F and G) in vitro need to be performed in Figure 3a, Figure 4b and Figure 4d.

We have included data for neutralization of all serovars D-G in the figures (Figure 2, Figure 3, and Figure 4).

2. In Figure 4b, serum from CTH522 exhibited strong *in vitro* neutralization capacity, however, the *in vivo* neutralization assay at day 7 post infection did not reflect *in vitro* results. The authors need to address it, meanwhile, what are the *in vivo* neutralization assay results at day 14, 21 and 28 post infection.

*The *in vivo* neutralization assay (Fig. 4c) is done to support our *in vitro* results in the HaK and the endocervix cell lines. Serum is heat-inactivated and incubated with EB and the suspensions are then inoculated intravaginally. The assay is performed to assess the immediate control of infection i.e. neutralizing chlamydial infectivity for mouse vaginal (cervix) epithelial cells and the primary measurement point is therefore day 3, supplemented with day 7 (as done by Su et. al. 1995). No additional antibodies are administered and the later outcome of the infection will depend on whether the infection was completely cleared at day 3. Once intracellular C.t. will multiply since no CTH522 specific T cells and no additional CTH522 specific antibodies from the blood will reach the genital tract. Followingly, we do not swab the mice for longer than PID7.*

3. In Figure 5c, do the mice culture negative at day 3 post infection remain negative though the whole culture period in Exp.1 and Exp.2?

No, the mice that culture negative at PID 3 do not remain negative through the whole culture period except for one mouse. Many of them control the infection for a long time. Out of the 10 mice that were culture negative at PID7, 6 mice were still culture negative at PID21.

4. In lines 221-223, authors stated “In mice with sustained high levels of IFU in the vaginal tract, throughout the investigated time period, the inflammatory response progressed into fluid filled hydrosalpinx at PID29 (Fig. 5e, panel 6). In Depl. + CTH522 IgG group, what is the percentage of infected mice with high IFU developing pathology in Oviduct/ovary?”

We have rewritten the sentence p12 | 229 to clarify:

“In contrast, mice that did not control the initial infection and had sustained high levels of IFU in the swabs throughout the investigated time period developed substantial oviduct inflammation with dense filling of debris in the lumen and visible EB inclusions (MOMP-positive staining) in both lumen and epithelial cells of the oviducts (Fig. 5e panel 3-5). A FACS analysis of uGT tissue at PID 29 demonstrated the presence of a high

percentages of neutrophils especially in mice receiving control antibodies, but also NK cells, macrophages, and dendritic cells (DC's) were present (Supplementary Fig. 9). Even severe pathology with fluid filled hydrosalpinx could be detected at PID29 (Fig. 5e panel 6)."

All mice with sustained high levels of IFU develops inflammation in the oviducts but not all mice have developed severe hydrosalpinx formations at this time point. It is however a matter of the time point selected for investigation (see supplementary Fig. 7).

5. In Figure 5e, authors need to characterize inflammatory cells in the tissue as T cells were depleted.

We agree. We have therefore included a FACS analysis of the uGT at PID29 in the two groups (Supplementary Fig. 9). The FACS analysis showed the presence of a high percentages of neutrophils especially in mice receiving control antibodies, but also NK cells, macrophages, and dendritic cells (DC's) were present.

6. In Figure 6f, what is the percentage of infected mice from CTH522 group developing pathology in Oviduct/ovary?

In fully immunocompetent mice few bacteria reach the oviduct/ovary and less pathology develops. This is one of the drawbacks of using a human C.t. strain in fully immunocompetent mice. We have not done enough histopathology scoring in those mice to give a percentage. As seen in figure 6g only 1 out of 36 mice had bacteria in the oviducts in the CTH522 vaccinated mice compared to 13 out of 36 in the control group. How many of these mice that eventually would have developed pathology at a later time point, we are not able to say with certainty. The level of bacteria in the upper genital tract of fully immunocompetent mice are much lower compared to the level in the T cell depleted model.

7. Native MOMP provide protection against infectious burden but not pathology in a nonhuman primate trachoma model has been previously reported (PMID: 19494332), which should be referenced.

We agree. Results from non-human primate models have been neglected in the manuscript and we have corrected that both in the introduction p3. l 42 where we have inserted the ref: (Kari et al, 2009) and in the discussion p18. l 378 : "In a non-human primate experiment with a native MOMP subunit vaccine, reduction in ocular shedding was observed, albeit no difference from control animals in progression of ocular disease (Kari et al, 2009)

Reviewer #3 (Remarks to the Author):

Immune signature of the Chlamydia vaccine CTH522/CAF1®01 translates from mouse to human and induces durable protection. Olsen et al.

General: This is a well written paper that describes the T cell and antibody immune responses to a MOMP peptomer vaccine adjuvanted with CAF01 after delivery in human females and mice. Confirmation of the ability of the vaccine to significantly reduce bacterial burden in female mice challenged with human serovar D of C. trachomatis is confirmed (previously published by this group) and additional data indicate reduced

ascension and protection from oviduct pathology in vaccinated mice. Data indicate similar T cell responses are induced in humans and mice after vaccination with CTH522/CAF01. Appropriate controls are included and sufficient numbers of mice and repeat experiments provide evidence of rigor and reproducibility of the results. The long-term protection studies are particularly impactful.

Statements related to the protective role of antibody should be deemphasized and qualified in the discussion. Many T cell depleted mice continue to develop upper genital tract infection and oviduct disease despite development of high levels of CTH522-specific antibody. Further, no studies were conducted that remove the effects of antibody and test for reduced protection in the absence of antibody, e.g., vaccination and challenge of B cell deficient mice, or adoptive transfer of immune T cells into RAG-deficient mice – such studies would complement the current data.

Although these additional studies are not required – since they were not performed, and human *C. trachomatis* inoculated into mice produces an infection of reduced intensity and duration compared to *C. muridarum* inoculated into mice due to host species-specific chlamydia-specific effects of interferon-gamma – the role of antibody may be exaggerated in this model. Since the infection is already attenuated using this *C. trachomatis* in female mice model – the ability of antibody to inhibit infection further may be accentuated.

Statements related to the protective role of antibody should be deemphasized and qualified in the discussion. Many T cell depleted mice continue to develop upper genital tract infection and oviduct disease despite development of high levels of CTH522-specific antibody. Further, no studies were conducted that remove the effects of antibody and test for reduced protection in the absence of antibody, e.g., vaccination and challenge of B cell deficient mice, or adoptive transfer of immune T cells into RAG-deficient mice – such studies would complement the current data.

We agree and have now made several additions to the discussion regarding the protective role of antibodies:

P17 l. 359: “Even though bacteria were coated with immune sera, there were still mice that failed to control the infection. Likewise, adoptively transferred CTH522 specific IgG failed to reduce the initial level of bacteria in the uGT of 20 out of 30 mice. Within the investigated time period (29 days) we did, however, see an overall lower burden in the uGT in the group receiving CTH522 antibodies compared to controls. This highlights that controlling chlamydial infections and targeting the bacteria’s biphasic life cycle, which has an intracellular and extracellular stage, requires an adaptive T cell response in addition to antibodies.”

p21 l 432. “Serum antibody levels decreased 11-fold after 6 month and 67-fold during the one-year period. This antibody trajectory likely means that over time the role of neutralizing antibodies in the immediate control of infection will decline, and instead rely on the memory T and B cell response (Fig. 7c). In fact, we measured a significant reduction in IFU levels already at PID 3, which became more pronounced at PID7 where the level of bacteria in the CTH522/CAF01 group was reduced from a median log₁₀ IFU of 4.1 (PID3) to 0.6 (PID7). We saw a recall antibody and CMI response following infection, slightly offset bacterial clearance, which could impact clearance of the infection and bacterial shedding.

Many of the suggested experiments have been performed using the Hirep construct which is the immunorepeat part of CTH522 (Olsen et al. 2015 and 2017)

Title: The title suggests the vaccine has been demonstrated to induce durable protection in humans. The title should be rewritten to better indicate what has been demonstrated in mice and what has been demonstrated in humans.

The title is rewritten to: Immune signature of Chlamydia vaccine CTH522/CAF®01 translates from mouse-to-human and induces durable protection in mice

Abstract: “Interestingly” – could be deleted – let the reader decide what is interesting.

Interestingly has been deleted

Introduction:

1. Fibrotic blockage of the tubal and resulting infertility or ectopic pregnancy – should be fibrotic blockage of the fallopian tubes or fibrotic blockage of oviducts.

2. Typos

– Overall, the outcomes of our comparative analysis of CTH522 induced immune responses in mice and humans are promising in relation to the protective efficacy of CTH522 as a Chlamydia vaccine for humans that is likely to be introduced alongside the HPV vaccine and therefore needs to be sustained for at least a decade.

CTH522 is a recombinant engineered version of the Major Outer Membrane Protein (MOMP) from C.t. MOMP and holds four highly variable domains VD1 to VD4, spaced between serovar conserved segments (CSI-CSV).

To localize and compare the breadth of the T cell epitope regions of CTH522, mouse splenocytes and human PBMCs were further stimulated...

Thank you to the reviewer for pointing this out. All the typos have been corrected.

Results:

1. Preincubation of chlamydia in high levels of antibody-containing serum for an hour is likely very different than antibody interactions with CT EBs in cervicovaginal mucus or male urethral secretions/ejaculate. This should be mentioned in the discussion as a limitation of the experiments in immunologically normal mice that examined the role of antibody. Should also remark that CT infection is compromised in mice due to host-species-specific immune evasion and thus the model used could overestimate protective effects of antibody (see above).

We agree and a section on the limitations of the mouse model and the limitation of the neutralization assays has been inserted p 21 | 446:

A limitation of the current study is the use of human *C.t.* serovars in the mouse model, as this does not fully reflect the pathogenesis and immune responses of human chlamydial infections. Especially, IFN- γ -mediated effector mechanisms leave *C.t.* more vulnerable in the mouse model, which means both T cell and antibody effector mechanisms could be overestimated (Perry 1999). Another limitation is to what extent the *in vivo* neutralization assay mimics the antibody interaction with *C.t.* in cervicovaginal mucus. We preincubated for 45 min. with a high level of antibody-containing serum, as done by Su et al. which informs to what extent we can block *C.t.* from infecting epithelial cells. Although antibody levels will be significantly lower in the GT following vaccination, it has been shown that antibodies, even in low concentrations, interact with the cervical mucin and entrap bacteria and viruses in the mucin mesh, which could slow down the interaction with the epithelial cells, as seen during HSV infection (Wang et al 2014).

2. The authors tend to overstate impact of antibody – In experiments that used T cell depleted mice - “Ten out of 30 mice (sum of both experiments) receiving CTH522 specific antibodies had a marked effect of the transferred antibodies, with no detectable bacteria at PID7.” The alternative– is that in 20 out of 30 mice minimal effect was observed – this should be reported. e.g., despite high levels of antibody – 20 of 30 mice continued to exhibit burden levels that were not significantly different than controls.

We fully agree and we have now made a more balanced discussion regarding the protective role of antibodies. Se inserts in discussion P17 l. 359:

“Even though bacteria were coated with immune sera, there were still mice that failed to control the infection. Likewise, adoptively transferred CTH522 specific IgG failed to reduce the initial level of bacteria in the IGT of 20 out of 30 mice. Within the investigated time period (29 days) we did, however, see an overall lower burden in the uGT in the group receiving CTH522 antibodies compared to controls. This highlights that controlling chlamydial infections and targeting the bacteria’s biphasic life cycle, which has an intracellular and extracellular stage, requires an adaptive T cell response in addition to antibodies”

Discussion:

1. The authors should discuss that the immunization failed to neutralize serovar H, I Ia, and J despite inclusion of conserved regions of MOMP in the vaccine construct.

We agree that this is a key point and have tried to clarify this p19 l 393- and by inserting new data as supplementary fig. 5

*Blocking of *C.t.* SvD, E, F, and G neutralization with a construct representing the VD4 region for all serovars D-G (Fig. 3c) demonstrated that the neutralizing epitopes in CTH522 were located to the VD4 region. We further demonstrated that CTH522 specific antibodies were able to neutralize serovars present in the B (SvD, E, B), B-related (SvF and SvG), and C-related complex (SvK) and to a lesser extent the C-complex serovars (SvH, I, Ia, J), except for SvA where we did see detectable levels of neutralization. This serovar restricted neutralization likely reflects to which degree the VD4 region is surface accessible in the individual serovars (Zhong et al 1990). Recently, we published a paper focusing on VD1 based vaccine constructs from SvA and SvJ demonstrating VD1 based neutralization of C and C-related complex serovars (Olsen et al 2021). In the present study, we*

*likewise demonstrated strong neutralization of C complex serovar (except for SvH) with an extVD1 immunorepeat based on Svla (extVD1^{la}*4) (Supplementary Fig. 5).*

- 2. Line 374 – “vaginal tract” – strange terminology- - Chlamydia don’t infect the vagina – they infect the exocervix and endocervix and ascend up. The antibodies are protecting from cervical infection.**
- 3. Otherwise – very nice balanced discussion of roles of CMI and antibody.**

Thank you to the reviewer for pointing this out, we have changed it accordingly and no longer use the terminology “Vaginal tract”.

REVIEWER COMMENTS

Reviewer #1 (Remarks to the Author):

No additional comments

Reviewer #2 (Remarks to the Author):

The revised manuscript is greatly improved, the authors addressed the points I had raised, In the revised version, there are couple of remaining points that need to be addressed.

1. In figure 2i, CTH522 specific serum can not provide cross neutralization against SvH, I, Ia , and J. Authors need to discuss how to improve CTH522' potential application.

2. In figure 4c and 5e, CTH522 specific serum provided in vivo neutralization at 1:2 and 1:4; the inclusion can be found in CTH522 serum treated group. These results suggest that the efficiency of neutralization antibodies produced by CTH522 needs to be improved. Authors need to discuss it.

Point by point response to reviewers' comments

Reviewer #1 (Remarks to the Author):

No additional comments

Reviewer #2 (Remarks to the Author):

The revised manuscript is greatly improved, the authors addressed the points I had raised, In the revised version, there are couple of remaining points that need to be addressed.

1. In figure 2i, CTH522 specific serum can not provide cross neutralization against SvH, I, Ia , and J. Authors need to discuss how to improve CTH522' potential application.

2. In figure 4c and 5e, CTH522 specific serum provided in vivo neutralization at 1:2 and 1:4; the inclusion can be found in CTH522 serum treated group. These results suggest that the efficiency of neutralization antibodies produced by CTH522 needs to be improved. Authors need to discuss it.

This is valid points and to address these comments we have inserted sections in the discussion (red) and moved the whole section further forward in the discussion:

Discussion p. 18 l. 366:

We further demonstrated that CTH522 specific antibodies were able to neutralize serovars present in the B (SvD, E, B), B-related (SvF and SvG), and C-related complex (SvK) and to a lesser extent the C complex serovars (SvH, I, Ia, J), except for SvA where we did see detectable levels of neutralization. This serovar restricted neutralization likely reflects to which degree the VD4 region is surface accessible in the individual serovars³⁴. **CTH522 will therefore target the majorities of circulating strains with neutralizing and surface recognizing antibodies in combination with a broad and cross-protective CMI response which potentially can cover all C.t. serovars.** Recently, we published a paper focusing on VD1 based vaccine constructs from SvA and SvJ demonstrating VD1 based neutralization of C and C-related complex serovars³⁵. In the present study, we likewise demonstrated strong neutralization of C complex serovar (except for SvH) with an extVD1 immuno-repeat based on SvIa (extVD1^{Ia*4}) (Supplementary Fig. 5). Our previous results on VD1 based vaccine construct and the present study agrees with what has previously been reported on MOMP specific monoclonal antibodies^{58, 59, 60, 61}. **As with the HPV vaccine, it is therefore a possible scenario that the CTH522 vaccine can be further improved along the way, when its efficacy in humans has been established (Phase IIb/III). The vaccine could e.g. be supplemented with an immuno-repeat based on VD1 regions from C complex serovars in order to further broaden the serovar coverage**

Discussion p.20 l. 412:

Despite the fact that CTH522 antibodies alone, either by directly coating the bacteria (in vivo neutralization, Fig. 4c) or when adoptively transferred to T cell depleted mice (Fig. 5 c,e) was

not sufficient to completely control the infection in all mice, the combined/synergistic response of antibodies and CMI in immunocompetent mice was highly protective. We show that vaccination with CTH522/CAF@01 significantly protect against infection in the lower genital tract (Fig. 6e) and against the ascending bacteria in the uterine horns and oviducts/ovaries (Fig. 6f). These mice contained both neutralizing antibodies in the GT (Fig. 6d) as well as cellular immunity (Fig. 6b and 7c).